# Can It Edit? Evaluating the Ability of Large Language Models to Follow Code Editing Instructions

**Federico Cassano**
Northeastern University
cassano.f@northeastern.edu

**Luisa Li**
Northeastern University

**Akul Sethi**
Northeastern University

**Noah Shinn**
Northeastern University

**Abby Brennan-Jones**
Wellesley College

**Jacob Ginesin**
Northeastern University

**Edward Berman**
Northeastern University

**George Chakhnashvili**
Northeastern University

**Anton Lozhkov**
HuggingFace

**Carolyn Jane Anderson**
Wellesley College

**Arjun Guha**
Northeastern University and Roblox
a.guha@northeastern.edu

## Abstract

A significant amount of research is focused on developing and evaluating large language models for a variety of code synthesis tasks. These include synthesizing code from natural language, synthesizing tests from code, and synthesizing explanations of code. In contrast, the behavior of instructional code editing with LLMs is understudied. These are tasks in which the model is provided a block of code and an instruction to modify the code. The editing instruction may ask for a feature to be added or removed, describe a bug and ask for a fix, or ask for a different kind of solution. We introduce a carefully crafted benchmark of code editing tasks and use it to evaluate several cutting edge LLMs. Our evaluation exposes a significant gap between the capabilities of state-of-the-art open and closed models. For example, even GPT-3.5-Turbo is better than the best open model at code editing tasks. We also introduce a new, carefully curated, permissively licensed training dataset of code editing tasks coupled with natural language instructions. Using this training dataset, we show that we can fine-tune open Code LLMs to significantly improve their code editing capabilities, closing the gap between open and closed models. All code, data, and models are available at https://github.com/nuprl/CanItEdit.

## 1 Introduction

Large language models of code (Code LLMs) are becoming an essential tool for software engineering practice and research. There has been significant research on synthesizing code from natural language instructions, but comparatively less attention has been given to code editing tasks. However, LLM users expect models to be capable of editing code. For example, the LMsys dataset of in-the-wild conversations with chatbots (Zheng et al., 2023) has 4,188 conversations containing code, and 831 (19%) of these involve code editing, where the user prompts the model to update code based on natural language instructions (Appendix E). In general, code editing with an LLM encompasses activities like feature addition or removal, bug fixing, and code refactoring (Zhang et al., 2023b; Moon et al., 2023; Shinn et al., 2023; Chen et al., 2023; Olausson et al., 2023; Jin et al., 2023).

The ability to edit code is also essential for a model to be useful for an AI-focused code editor such as Cursor (Cursor, 2023), Copilot Chat (Copilot, 2023), or ChatGPT Advanced

**Instruction:** Edit the C4 class and its methods to represent the C8 group instead.

```diff
-class C4(nn.Module):
+class C8(nn.Module):
-    """Represents the cyclic group C4,
+    """Represents the cyclic group C8,
        where each element represents a discrete rotation."""

     def __init__(self):
         super().__init__()

     def size(self):
         """Outputs the size of this group."""
-        return 4
+        return 8

     def elements(self):
         """Returns all the elements of this group"""
-        return torch.tensor([0., np.pi/2, np.pi, 3*np.pi/2])
+        d = np.pi / 4
+        return torch.tensor([0., d, d*2, d*3, d*4, d*5, d*6, d*7])
```

Figure 1: An abbreviated example of a code editing task from the CANITEDIT dataset (Figure 9 presents the full example). The model is tasked with editing the `C4` group to represent `C8` instead. The model is expected to infer the after code segment from the instruction and the before code segment, as shown in the inferred code diff.

Data Analysis (ADA) (OpenAI, 2023a). Cursor and Copilot Chat facilitate edits with human-written instructions. In contrast, ADA uses both human-written instructions and model-generated *reflections* (Shinn et al., 2023; Fan et al., 2023; Phung et al., 2023) to extend and edit code. This approach represents a step towards AI-driven code assistance. In both scenarios, *instructional code editing* is employed, which we define as a function $M(c, I) \rightarrow c'$, where $c$ is the original code, $I$ is the instruction, and $c'$ is the modified code. Figure 1 illustrates this process, showing how the model edits a code segment from a given instruction.

Model-generated reflections and human-written instructions both describe desired code changes. However, they differ in the level of detail: reflections, usually more detailed, are generated by a model with access to the code, offering richer context and potentially a strategic plan for code modifications. In contrast, human-written instructions are typically shorter and less detailed but may express the true user's intent more clearly. We refer to these as *descriptive* and *lazy* instructions, respectively. We thoroughly analyze examples of such instructions in Appendix E.

In this work, we introduce CANITEDIT, a novel dataset comprising 105 hand-crafted instructional code editing problems, featuring both descriptive and lazy instructions and an extensive hidden test suite. Designed to assess a model's proficiency in handling diverse code editing scenarios, CANITEDIT serves as a benchmark for evaluating state-of-the-art Code LLMs in instructional code editing. Our evaluation focuses on measuring the accuracy of a given model's ability to write correct code modifications without introducing unnecessary code. We conduct comprehensive assessments of closed and open models, revealing significant performance disparities between the leading closed and open models (§5). To help address this gap, we propose a training dataset and methodology for code editing. Our findings demonstrate that fine-tuning open Code LLMs on this dataset can significantly enhance code editing performance (§4).

To summarize, we make four main contributions: (1) We introduce CANITEDIT, a dataset of instructional code editing problems, designed to evaluate the code editing capabilities

of large language models (§3). (2) We propose a novel metric, *ExcessCode*, for quantifying the typical volume of unused code produced by a model when generating the correct code edits (§5.1). (3) We perform a thorough evaluation of the latest Code LLMs in the context of code editing, providing insights into their current capabilities (§5). (4) Finally, we present a specially tailored training dataset for code editing, along with an effective training methodology, demonstrating significantly enhanced code editing performance through fine-tuning models of varying sizes (§4).

## 2 Related Work

**Instruction-following Language Models.** Correctly prompting an LLM is crucial for it to perform a desired task. There are multiple methods for *instruction tuning* LLMs to better adhere to natural language instructions. One method involves employing human annotators to create sample instructions and provide feedback on numerous model outputs (Ouyang et al., 2022; Köpf et al., 2023). This method is costly and demands substantial resources. An alternative, cost-effective method is to use an LLM to *self-instruct*, generating instructions from a smaller set of human-written seed instructions (Wang et al., 2023). These methods have been applied to generate datasets for instruction-tuning Code LLMs (Chaudhary, 2023; Luo et al., 2023). Specific to code generation, another strategy to instruction-tune an LLM is to use commit messages as instructions (Muennighoff et al., 2023). In this paper, we use commit messages as instructions for code editing. Our results demonstrate that while instruction-tuned models can edit code, they are not as effective as models that we explicitly train for this task (§5).

**Code Generation Benchmarks.** Several benchmarks exist that test a model's code generation ability. HumanEval and MBPP are two prominent benchmarks for evaluating LLMs in Python programming (Chen et al., 2021; Austin et al., 2021). MultiPL-E expands these benchmarks to 18+ additional programming languages (Cassano et al., 2023b). These benchmarks assess model-generated candidate completions against a series of human-authored unit tests. EvalPlus (Liu et al., 2023) utilizes mutation testing to expand the test suites of the Python benchmarks. EvoEval (Xia et al., 2024) addresses limitations in existing code generation benchmarks by evolving them into diverse domains. It reveals significant performance drops in LLMs compared to standard benchmarks, highlighting the need for more comprehensive evaluation methods. All of these benchmarks utilize the *pass@k* metric, which measures the likelihood of the model generating a completion that passes all of the tests in *k* tries; we also adopt this metric in our evaluation (§5.1). However, these benchmarks are limited to the evaluation of a model's ability to generate a single function from a natural language description and do not assess code editing capabilities. HumanEvalPack (Muennighoff et al., 2023) is a benchmark designed for evaluating LLMs across various single-function code generation tasks, such as synthesis, code explanation, and bug fixing. Specifically, HumanEvalFix, a bug-fixing variant of HumanEvalPack, is extensively used for assessing the models' capabilities in code refinement (Moon et al., 2023; Muennighoff et al., 2023). However, the instruction is fixed for every problem. SWE-Bench (Jimenez et al., 2023) evaluates LLMs across varied programming tasks including planning, retrieval, and code editing. Our work concentrates specifically on code editing tasks, aiming to more precisely guide model development. Unlike SWE-Bench, which sources its problems from GitHub PRs and issues, our benchmark is handcrafted, reducing contamination risks as seen with models like StarCoder and StarCoder2, which are trained extensively on GitHub data (Li et al., 2023b; Lozhkov et al., 2024).

**Code Editing Using Large Language Models.** Previous studies on code editing with LLMs have predominantly focused on bug fixing (Zhang et al., 2023b; Moon et al., 2023; Shinn et al., 2023; Chen et al., 2023; Olausson et al., 2023; Jin et al., 2023; Joshi et al., 2023; Wei et al., 2023; Li et al., 2022), a specific subset of code editing; fill-in-the-middle code completion (Bavarian et al., 2022; Fried et al., 2023; Yee & Guha, 2023; Roziere et al., 2023; Guo et al., 2024a; Zhang et al., 2023a), an inference strategy that requires specific insert locations; and intrinsic code editing (Li et al., 2023a; Gupta et al., 2023), which involves editing code without a specified instruction, exerting the model's ability to intrinsically ascertain the desired code changes. Recently, LLMs have progressed in code editing guided by natural

| CanItEdit Dataset Statistics | |
| --- | --- |
| Total Tasks | 105 (35/35/35) |
| Total Problems | 210 (70/70/70) |
| **Topics** | |
| Data Structures & Algorithms | 39 |
| Language Processing | 21 |
| Mathematics | 25 |
| Data Science | 10 |
| Miscellaneous | 10 |
| Problems With Library Usage | 22 |
| **Code Segment** | **Mean $\pm$ Std. Dev.** |
| Mean Lines (Before\|After) | 42.5 $\pm$ 33.9 \| 49.8 $\pm$ 36.6 |
| Levenshtein Distance | 302.1 $\pm$ 339.6 |
| Combined Mean Lines | 92.3 $\pm$ 69.9 |
| Combined Mean Tokens | 865.3 $\pm$ 639.7 |
| Combined Max Tokens | 3,583 |
| **Instruction** | **Mean $\pm$ Std. Dev.** |
| Mean Tokens (Descriptive\|Lazy) | 81.7 $\pm$ 50.4 \| 35.6 $\pm$ 30.6 |

Table 1: Dataset statistics for CANITEDIT.

language without specific edit locations (Hu et al., 2023; Li et al., 2023b; Muennighoff et al., 2023). However, this advancement lacks benchmark evaluations to effectively measure the models' code editing skills. Notably, StarCoder (Li et al., 2023b), the first LLM trained on an extensive dataset of commits using the format <before><commit message><after>, we have shown enhanced code editing capabilities (§5). Before this study, StarCoder's practical code editing performance had not been assessed. StarCoder2 has replaced commits with pull requests and issues, which typically include more natural language (Lozhkov et al., 2024). CodeEditorBench (Guo et al., 2024b) evaluates code editing using competitive programming problems. Unlike CANITEDIT, it uses fixed prompts and lacks diverse topics and dual instruction sets. The recent introduction of InstructCoder (Hu et al., 2023), a model explicitly trained and evaluated for instructional code editing, marks a significant step towards code editing with LLMs. However, its evaluation involved GPT-4-generated (OpenAI, 2023b) and human-provided labels, which raises issues regarding reproducibility and comparability in future research. Moreover, the model has not been publicly released, prohibiting us from evaluating it on our benchmark.

## 3 The CANITEDIT Dataset

**Benchmark Overview** CANITEDIT is a dataset comprising 105 meticulously constructed Python code editing challenges. Each problem includes the input code segment (*before*), the expected code segment (*after*), the two types of natural language instructions (descriptive and lazy), and a hidden test suite. These challenges span a broad spectrum of computer science domains, such as data structures, algorithms, mathematics, language processing, and game programming, requiring knowledge of popular external Python libraries like NumPy, Pandas, PyTorch, and others. Table 1 presents general dataset statistics for CANITEDIT.

Following Swanson (1976) and follow-up work (Levin & Yehudai, 2017), we classify code editing tasks into three distinct categories based on their primary goal: a *corrective* edit fixes errors, a *perfective* edit enhances existing features, and an *adaptive* edit meets new requirements. We have 35 problems per category for an even distribution across the different types of code changes. The dual instruction formats test the models' ability to execute tasks based on the provided context: descriptive instructions provide comprehensive details for explicit guidance, while lazy instructions offer minimal direction, challenging the model to infer

the required actions. Both instructions should lead to an equivalent after segment. Descriptive instructions serve to replicate situations where users provide specific specifications or another model outlines a plan. In contrast, lazy instructions resemble typical user queries for LLMs in code generation. As each problem has two distinct instructions, the dataset effectively contains 210 problems. We showcase examples from CANITEDIT in Appendix C.

**Dataset Creation**   To manually construct CANITEDIT, we assembled a team of eight experienced Python programmers, each with different domain expertise, and appointed one as the lead. The team, recruited through academic networks, included researchers and engineers from various fields who use Python extensively. Our objective was to fill each change category (corrective, perfective, and adaptive) with 35 problems, for a total of 105 problems. The change category is determined by the type of edit required to transform the 'before' code segment into an 'after' code segment that passes the hidden test suite.

Before starting, we provided the team with verbal and written guidance, a standard template, and an example problem. They were instructed to begin by writing a brief description of the problem and the applied changes for review and refinement by the lead. Next, they were tasked to write the 'before' code segment and hidden test suite, followed by the 'after' code segment, along with both lazy and descriptive instructions. The lead initially reviewed all problems in development, and they were additionally reviewed by the entire team in weekly meetings. Upon completion of a problem, the lead generated sample completions to ensure that the failures and successes were reasonable and consistent with the problem's intent.

The team dedicated significant effort to developing comprehensive test suites for each problem, incorporating a variety of testing techniques such as unit tests, property-based testing, mocking, fuzzing, and integration tests. These suites were designed to rigorously evaluate whether the 'after' segment met the problem requirements while ensuring the 'before' code did not. To confirm the completeness and correctness of the test suites, we created an automated verification pipeline that ensured 100% line coverage and that the suite passed all tests with the 'after' code while failing at least one with the 'before' code. The team also manually reviewed the tests to ensure correctness and completeness.

Annotators were encouraged to create problems of varying difficulty and were assigned specific topics and change types. The lead provided example problems and guidance on writing unambiguous instructions to ensure fair model evaluation. All instructions and several model completions were reviewed by the project lead to maintain consistency and clarity across the dataset.

## 4   Fine-Tuning

We describe our approach to fine-tuning Code LLMs for code editing tasks, focusing on the DeepSeekCoder-Base family (Guo et al., 2024a), a variant of CodeLlama (Roziere et al., 2023) trained on 2 trillion tokens of GitHub code and natural language, using StarCoder's filtering rules (Li et al., 2023b). These models, top-performing in code generation and open-access under a permissive license, show robust performance on CANITEDIT without specific training for instructional tasks (§5).

For our ablation studies, we focus on the model with 6.7 billion parameters, which offers an ideal balance between size and performance. This allows us to extrapolate results to larger models with more parameters without the need for extensive training. Following the most performant training strategy identified, we also fine-tune the 1.3b and 33b models to evaluate the impact of model size on code editing performance. Our fine-tuned models are referred to as EDITCODER. These models have been full-parameter fine-tuned by calculating the loss on only the 'after' code segment. Appendix B provides further details and experiments on the training process.

We experiment with two training datasets we gathered: EditPackFT and Commits2023FT, which we describe below. Table 2 presents the statistics for these datasets.

| Dataset Statistics | | |
|---|---|---|
| | **EditPackFT** | **Commits2023FT** |
| Total Commits | 22,602 | 24,129 |
| Unique Initial Verbs | 184 | 199 |
| Code Segments (Mean $\pm$ Std. Dev.) | | |
| Lines of Code | $29.2 \pm 13.7$ | $119.3 \pm 75.9$ |
| Levenshtein Distance | $197.1 \pm 260.6$ | $406.6 \pm 631.2$ |
| Commit Messages (Mean $\pm$ Std. Dev.) | | |
| Tokens | $10.1 \pm 4.6$ | $23.1 \pm 35.2$ |

Table 2: Training dataset statistics for EditPackFT and Commits2023FT

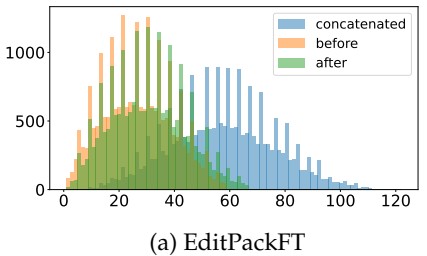

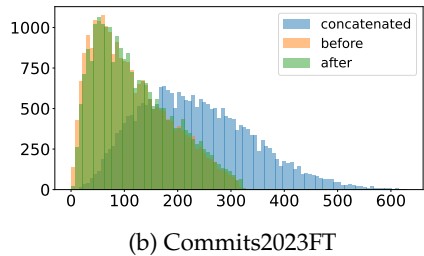

(a) EditPackFT          (b) Commits2023FT

Figure 2: The distribution of the number of lines in the 'before' and 'after' code segments in the EditPackFT and Commits2023FT datasets. The 99th percentile is removed for clarity.

**EditPackFT**  We created the EditPackFT dataset by further filtering the Python split of the CommitPackFT dataset (Muennighoff et al., 2023). CommitPack is an extensive dataset comprising 4TB of permissively licensed commits from a 2016 GitHub snapshot across various programming languages. CommitPackFT is a subset of CommitPack, filtered to be amenable for instruction-tuning Code LLMs. The primary criterion for CommitPackFT's selection involved retaining commits whose messages begin with an imperative verb, mirroring the typical structure of natural language instructions. We apply a series of additional filtering steps, which make the dataset more suitable for code editing. We remove any item that pass any of the following predicates:

1. The presence of an empty 'before' or 'after' code segment, disregarding whitespace.
2. No change detected in the 'before' and 'after' code segments.
3. The inclusion of the words *TODO*, *FIXME*, or *BUG* in the 'after' code segment, which signals an incomplete commit.
4. Incorrect parsing of the 'after' code using the Python `ast` module.

Originally, the dataset contained 56,025 commits, and after applying the filtering steps, we are left with 22,602. As shown by Figure 2a and Table 2, the mean number of lines in the 'before' and 'after' code segments is 29.2. The mean Levenshtein distance between the 'before' and 'after' code segments is 197.1 characters, indicating that the changes are small relative to CANITEDIT's 302.1 characters. We also analyze the distribution of the commit message lengths, and find that the mean token count is 10.1, which is quite low compared to CANITEDIT's 81.7 tokens for the descriptive and 35.6 tokens for the lazy instructions. We further analyzed the original CommitPackFT dataset to ensure that these findings weren't artifacts of our filtering strategy: the full dataset has similar statistics.

**Commits2023**  To address the limitations of EditPackFT, we developed the Commits2023FT dataset. This dataset is filtered from 416,792 Python file changes gathered from commits in permissively licensed GitHub repositories. We name the unfiltered dataset Commits2023. Our objective is to create a dataset akin to CommitPackFT, but with more recent data and a

more diverse example length distribution. We employed the same filters on this dataset as used for EditPackFT, and also applied the initial filters from CommitPackFT, which includes only commits with messages that start with an imperative verb followed by at least a noun.

Additionally, we only retain one file from multi-file commits to avoid exact duplicate commit messages in our dataset. After this filtering process, we obtain a dataset comprising 24,129 Python file changes. Figure 2b presents a broader distribution of the number of lines in the 'before' and 'after' code segments, with an average of 119.3. We also observe a much larger change distribution, with a mean Levenshtein distance of 406.6, signaling that this is not only a dataset with larger code segments, but also contains more varied changes. Figure 3 illustrates a sunburst plot of the most frequent initial verbs in the commit messages of Commits2023FT, along with their corresponding root nouns. This set of verbs is slightly more varied than those in Edit-PackFT, featuring 199 unique verbs in comparison to 184. Furthermore, the token count distribution of the commit messages is twice as high and much more varied than that of EditPackFT, with a mean of 23.1 and a standard deviation of 35.2.

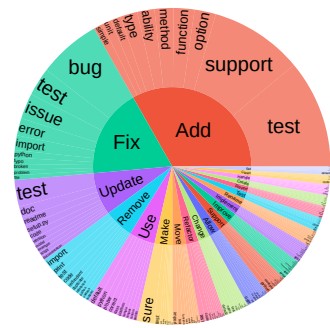

Figure 3: Sunburst plot of the top 20 most frequent initial verbs, with their corresponding top 10 root nouns, in the commit messages of Commits2023FT.

**Ablation Datasets**   For ablation analysis, we generated two additional datasets: Commits2023Raw25k and Commits2023FT+EditPackFT. Commits2023Raw25k consists of a random selection of 25,000 commits from Commits2023. We use this dataset to assess the impact of the filtering process on the final dataset. Commits2023FT+EditPackFT represents the combined dataset of Commits2023FT and EditPackFT. We find that the combination of Commits2023FT and EditPackFT yields the best results by a significant margin (§5.2), and thus we train our final models on this dataset. We believe that these results are due to the increased amount of data and the expanded length distributions.

## 5   Evaluation

In this section, we evaluate the performance of various open and closed models on the CANITEDIT benchmark, as well our fine-tuned models.

**Evaluation Tools and Hyperparameters**   We run the open-access models using Hugging-Face Transformers (Wolf et al., 2020) and vLLM (Kwon et al., 2023). We use the following hyperparameters for all inference experiments: 2048 maximum new tokens, temperature 0.2, and top-$p$ sampling cutoff of 0.95. Following Cassano et al. (2023b), we sample 20 completions for each problem. We run all tests in a Docker container to mitigate the risk of malicious code execution.

**Models Evaluated**   We evaluate several state-of-the-art models of varying sizes, fine-tuning some of them to build EDITCODER. We group the models into two categories: **open**, models which we have access to their weights, and **closed**, models which we do not. We prompt each model with their recommended prompt template. The specific templates used appear in Appendix A.6. The full list of models and their sizes appears in Table 3.

### 5.1   Evaluation Metrics

We employed two metrics to assess model performance: *pass@k* assesses functional correctness, and *ExcessCode* assesses conciseness and precision of code edits.

- *pass@k* is the likelihood that at least one successful edit was made from *k* attempts, as assessed by the test suite. In this section, we show results only for *pass@1*, and

| Model | | Descriptive | | Lazy | |
|---|---|---|---|---|---|
| Name | Size | *pass@1* | *ExcessCode* | *pass@1* | *ExcessCode* |
| Closed Models | | | | | |
| GPT-4 | — | **63.33** | $0.15 \pm 0.09$ | **51.95** | $0.14 \pm 0.10$ |
| GPT-3.5-Turbo | — | 48.14 | $0.47 \pm 0.34$ | 42.71 | $0.00 \pm 0.00$ |
| Open Models | | | | | |
| CodeLlama-Instruct | 70b | 45.05 | $0.28 \pm 0.15$ | 37.52 | $0.02 \pm 0.02$ |
| Mixtral-Instruct | 8x7b | 30.10 | $0.40 \pm 0.16$ | 24.90 | $0.01 \pm 0.01$ |
| EDITCODER | 33b | **55.90** | $0.33 \pm 0.21$ | **42.33** | $0.27 \pm 0.24$ |
| DeepSeekCoder-Instruct | 33b | 49.78 | $0.36 \pm 0.24$ | 38.94 | $0.51 \pm 0.34$ |
| DeepSeekCoder-Base | 33b | 47.71 | $0.53 \pm 0.24$ | 34.71 | $0.62 \pm 0.41$ |
| CodeLlama-Instruct | 34b | 30.63 | $0.33 \pm 0.21$ | 24.15 | $0.18 \pm 0.14$ |
| StarCoder2 | 15b | 41.95 | $0.36 \pm 0.20$ | 31.48 | $0.04 \pm 0.04$ |
| StarCoder | 15b | 37.10 | $0.56 \pm 0.28$ | 27.62 | $0.42 \pm 0.34$ |
| OctoCoder | 15b | 34.43 | $0.12 \pm 0.07$ | 25.95 | $0.07 \pm 0.07$ |
| CodeLlama-Instruct | 13b | 26.90 | $0.90 \pm 0.68$ | 16.89 | $0.42 \pm 0.41$ |
| EDITCODER | 6.7b | 48.33 | $0.36 \pm 0.17$ | 39.29 | $0.32 \pm 0.25$ |
| DeepSeekCoder-Instruct | 6.7b | 41.03 | $0.13 \pm 0.06$ | 31.65 | $0.22 \pm 0.12$ |
| DeepSeekCoder-Base | 6.7b | 32.62 | $1.01 \pm 0.42$ | 27.76 | $1.25 \pm 0.98$ |
| CodeLlama-Instruct | 7b | 32.83 | $0.31 \pm 0.15$ | 23.49 | $0.36 \pm 0.26$ |
| EDITCODER | 1.3b | 26.67 | $0.14 \pm 0.09$ | 21.43 | $0.20 \pm 0.12$ |
| DeepSeekCoder-Instruct | 1.3b | 26.22 | $0.32 \pm 0.18$ | 17.27 | $0.32 \pm 0.13$ |
| DeepSeekCoder-Base | 1.3b | 17.90 | $0.69 \pm 0.42$ | 11.76 | $2.79 \pm 2.29$ |

Table 3: Evaluation results of close and open-access models on CANITEDIT. We report the *pass@1* and *ExcessCode* metrics for both the descriptive and lazy prompts as well as the size of the model if available.

> evaluation results for *pass@10* and *pass@100* with higher temperatures can be found in Appendix A.2.

- *ExcessCode* evaluates the presence of unnecessary code changes, as indicated by the fraction of *changed lines* not covered by the test suite. We calculate this metric by averaging the mean line coverage for passing completions across all problems, omitting those with no successful completions. The Python code used to calculate this metric is found at Appendix A.1. We additionally report the standard error of the mean for this metric.

## 5.2   Results with Existing Models

We draw several conclusions from the full results in Table 3.

**Closed source models outperform open source models.** Our evaluation indicates a significant performance disparity between open and closed models. GPT-4, despite not being primarily trained on code-related tasks but general instruction-following, surpasses DeepSeekCoder-Instruct 33b – the leading open source model – by an average of 13% in *pass@1* for both descriptive and lazy tasks. DeepSeekCoder-Instruct stands out as the only open model exceeding any closed model's performance, surpassing GPT-3.5-Turbo for descriptive prompts.

DeepSeekCoder-Instruct utilizes an undisclosed instruction-tuning dataset, therefore direct comparisons with other open models may not be entirely fair. In contrast, Mixtral-Instruct (Jiang et al., 2024), comparable to OpenAI models in its general instruction-following training focus, significantly lags in performance against both closed models and open models specialized in code generation tasks. Lastly, CodeLlama-Instruct-70b, ranked second in

instruction-following capabilities, was developed using a dataset of examples generated by Llama 2 70b with a larger focus on code generation tasks. This may explain its superior performance compared to Mixtral-Instruct.

**Descriptive prompts yield better performance than lazy prompts.** Descriptive prompts result in a 8.68 absolute increase on average in pass@1 compared to lazy prompts, which may be the result of more detailed information and additional pointers in descriptive problems. Lazy instructions generally lead to lower *ExcessCode* for larger models. This may be indicative of lazy instructions introducing less noise in the task, as there are less tokens to attend to in the prompt given.

**Larger models perform better than smaller models.** Model size correlates positively with *pass@1*, and negatively with *ExcessCode*, indicating that larger models are more adept at precise edits to code. This pattern is most clearly seen in the evaluation results of DeepSeekCoder-Base, StarCoderBase, and StarCoder2, where a steady increase in performance is seen along with an increase in model size (Appendix A.5).

**Models pre-trained on commits are better at code editing.** Among open models, StarCoder is pre-trained on GitHub commits, while StarCoder2 focuses on GitHub issues. StarCoder outperforms similar-sized DeepSeek and CodeLlama models in our benchmark, despite their superior code generation capabilities. OctoCoder, a StarCoder-based model fine-tuned on instructions (Muennighoff et al., 2023), shows lower *pass@1* performance, suggesting instruction fine-tuning on commit-based models may reduce code editing efficacy. Conversely, StarCoder2, exchanging commits with issues in its training, improves in editing tasks with larger models (Appendix A.5). This may be attributed to extended training and modern architecture rather than data source shift.

## 5.3 Results after Fine-Tuning on Commits

| Training Dataset | | | Descriptive | | Lazy | |
|---|---|---|---|---|---|---|
| Name | #Tokens | #Items | *pass@1* | *EC* | *pass@1* | *EC* |
| Commits2023FT+EditPackFT | 74M | 46,274 | **48.33** | 0.36 | **39.29** | 0.32 |
| Commits2023FT | 62M | 24,129 | 46.57 | 0.42 | 37.19 | 0.22 |
| Commits2023Raw25k | 62M | 25,000 | 42.57 | 0.41 | 35.14 | 0.17 |
| EditPackFT | 12M | 22,602 | 47.14 | 0.40 | 36.05 | 0.07 |

Table 4: Ablation results of training DeepSeekCoder-6.7b-Base on different datasets and evaluating on CANITEDIT. We show the total number of tokens and items in each dataset, as well as the *pass@1* and *ExcessCode* (EC) metrics for both the descriptive and lazy prompts. The reported sizes of the datasets are after deduplication.

In addition to evaluating existing open models, we also fine-tuned pre-trained DeepSeek models (§4) to build EDITCODER, which we now evaluate.

**Optimal Dataset: Commits2023FT+EditPackFT.** In finding the best training dataset for DeepSeekCoder-6.7b-Base, the base model for EDITCODER, various ablation datasets were tested. Results in Table 4 show Commits2023FT+EditPackFT is the top performer for both descriptive and lazy instructions. The dataset's larger size and diverse data types, including varied commits, edits, and instructions, likely contribute to its superior performance.

**Fine-tuning on open commits can significantly improve code editing performance.** EDITCODER-33b surpasses all open models in *pass@1* for both descriptive and lazy instructions types, showing an overall 10.7% increase in *pass@1* and a notable decrease in *ExcessCode* compared to its base model, DeepSeekCoder-Base-33b. Additionally, we see a substantial increase in *pass@1* for every iteration of EDITCODER over its corresponding base model, with the largest improvement being a 45.1% increase at 6.7b.

Both EDITCODER-33b and EDITCODER-6.7b outperform GPT-3.5-Turbo in *pass@1* for descriptive instructions, with EDITCODER-33b also matching GPT-3.5-Turbo in for lazy ones. In higher temperature scenarios (Appendix A.2), EDITCODER-33b beats GPT-3.5-Turbo in

both instruction types for *pass@10* and *pass@100*, and even surpasses GPT-4 in *pass@100* for descriptive instructions. Analysis in Appendix A.3 shows EDITCODER excels in corrective changes but is less effective in perfective changes. Further analysis in Appendix A.4 shows that EDITCODER-33b outperforms all models, including GPT-4, in simple single-function bug fixes, and retains synthesis capabilities. This demonstrates the effectiveness of targeted fine-tuning on code editing datasets, addressing the distinct needs of instructional code editing compared to general code generation.

## 6   Conclusion

We present CANITEDIT, a benchmark designed to assess the instructional code editing skills of Code LLMs. It includes 105 hand-written code editing problems, each accompanied by dual natural language instructions: a "lazy" instruction that a human may write, and a "descriptive" instruction that may be generated by an agent revising code in a loop. Each problem has a comprehensive test suite. We evaluate contemporary state-of-the-art Code LLMs and reveal a significant gap between closed and open models. We also demonstrate that fine-tuning with a custom dataset and training methodology can significantly improve code editing capabilities across various model sizes. Our work provides a foundation for evaluating future enhancements in instructional code editing for Code LLMs, offering valuable tools and insights for AI-based software development research and practice.

**Limitations**   We evaluated LLMs in reproducing the entire 'after' code segment, which may not be the most token-efficient method. A potentially more efficient strategy would involve generating a list of specific changes to be applied to the 'before' code segment. Furthermore, our study does not explore varying prompt formats. Instead, we have adopted a format consistent with that used by other models (Li et al., 2023b). Another limitation is the size of our final training dataset, which is relatively modest. We have not investigated the potential benefits of utilizing larger datasets, which could notably enhance performance, particularly with larger models. Our work only targets Python. Similar results may be possible for other high-resource programming languages, but low-resource languages may require additional effort (Cassano et al., 2023a). We identify these areas as opportunities for future work.

## Acknowledgements

This work was partially supported by the United States National Science Foundation (Awards SHF-2052696 and SES-2326174). Federico Cassano was affiliated with Roblox for a significant part of his work on this project. We thank Northeastern Research Computing and Joydeep Biswas for providing computing resources.

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

# A  Additional Evaluation Details

In this section, we provide several additional details about our evaluation. We provide the following additional details:

1. A Python implementation for computing the *ExcessCode* metric in A.1.
2. Results of our evaluation at higher sampling parameters in A.2.
3. A deeper analysis of our results per change type in A.3.
4. An evaluation of EDITCODER on HumanEvalPack in A.4.
5. A deeper comparison of the first and second versions of StarCoder in A.5.
6. Each prompt format used in our evaluation in A.6.

**OpenAI Model Versions**  For our evaluation we use the following versions of the OpenAI models, which at the time of writing were the latest stable versions:

- GPT-4: `gpt-4-0613`
- GPT-3.5-Turbo: `gpt-3.5-turbo-0125`

## A.1  Computing *ExcessCode*

```python
def excess_code(before: str, after: str, lines_missing: int):
    """
    Compute the ExcessCode score for a single code edit completion.
    Args:
        before: The original code segment.
        after: The modified code segment.
        lines_missing: The number of lines with missing code coverage.
    Returns:
        The computed ExcessCode score.
    """
    import difflib
    differ = difflib.Differ()
    before_lines = before.splitlines()
    after_lines = after.splitlines()
    lines_changed = len(differ.compare(before_lines, after_lines))
    return lines_missing / lines_changed
```

Listing 1: A Python implementation for computing the *ExcessCode* metric

We provide a simple Python implementation for computing the *ExcessCode* metric in Listing 1. The function takes in as input the original code segment, the modified code segment, and the number of lines with missing code coverage. The lines with missing code coverage are computed using a code coverage tool, in our case, Coverage.py (Batchelder & Contributors to Coverage.py). For Coverage.py, the number of lines with missing code coverage can be obtained by running the command `coverage report -m`.

## A.2  Results At Higher Sampling Parameters

In this section, we evaluate the performance of various models under higher sampling parameters compared to those used in our main evaluation (§5).

**Standard Sampling Parameters**  In §5, we assessed several models on CANITEDIT using standard parameters: temperature of 0.2, top-$p$ of 0.95, and 20 samples. These parameters, often used in code generation tasks (Chen et al., 2021; Cassano et al., 2023b;a; Muennighoff et al., 2023; Li et al., 2023b; Lozhkov et al., 2024), balance the selection of higher probability

| Model | | Metrics | | | |
|---|---|---|---|---|---|
| Name | Size | *pass@1* | *pass@10* | *pass@100* | *ExcessCode* |
| Descriptive | | | | | |
| GPT-4 | — | **63.67** | **73.67** | 80.00 | 0.18 ± 0.09 |
| GPT-3.5-Turbo | — | 47.88 | 61.67 | 71.43 | 0.33 ± 0.21 |
| EDITCODER | 33b | 54.36 | 73.15 | **81.90** | 0.81 ± 0.38 |
| CodeLlama-Instruct | 34b | 28.50 | 52.00 | 64.76 | 0.38 ± 0.16 |
| StarCoder2 | 15b | 40.06 | 62.12 | 71.43 | 0.47 ± 0.19 |
| StarCoder | 15b | 33.31 | 59.80 | 70.48 | 0.76 ± 0.25 |
| CodeLlama-Instruct | 13b | 25.01 | 50.04 | 62.86 | 0.47 ± 0.29 |
| EDITCODER | 6.7b | 46.31 | 60.24 | 70.48 | 0.42 ± 0.23 |
| CodeLlama-Instruct | 7b | 29.93 | 52.86 | 65.71 | 0.76 ± 0.39 |
| EDITCODER | 1.3b | 26.29 | 40.22 | 47.62 | 0.50 ± 0.23 |
| Lazy | | | | | |
| GPT-4 | — | **52.55** | **64.56** | **71.43** | 0.12 ± 0.09 |
| GPT-3.5-Turbo | — | 40.93 | 53.33 | 60.95 | 0.29 ± 0.27 |
| EDITCODER | 33b | 40.34 | 58.63 | 68.57 | 0.46 ± 0.20 |
| CodeLlama-Instruct | 34b | 21.10 | 42.69 | 56.19 | 0.19 ± 0.08 |
| StarCoder2 | 15b | 30.23 | 48.82 | 59.05 | 0.17 ± 0.07 |
| StarCoder | 15b | 24.75 | 50.23 | 65.71 | 0.68 ± 0.25 |
| CodeLlama-Instruct | 13b | 16.67 | 38.52 | 58.10 | 0.67 ± 0.44 |
| EDITCODER | 6.7b | 37.74 | 51.16 | 57.14 | 0.33 ± 0.15 |
| CodeLlama-Instruct | 7b | 20.85 | 41.10 | 54.29 | 0.11 ± 0.06 |
| EDITCODER | 1.3b | 20.20 | 32.70 | 39.05 | 1.47 ± 1.20 |

Table 5: Evaluation results of models on CANITEDIT at higher sampling parameters. We report the *pass@1*, *pass@10*, and *pass@100* metrics for both the descriptive and lazy prompts, as well as the *ExcessCode* metric. The size of the model is reported if available.

tokens while allowing for sampling of lower probability tokens at conservative levels, addressing surface form competition (Holtzman et al., 2021), making it typically more effective than greedy decoding (Chen et al., 2021).

**Higher Sampling Parameters**    For this evaluation, we increased the sampling parameters to assess the models' robustness under more diverse generation conditions. Following Chen et al. (2021), we adopted a temperature of 0.8, top$-p$ of 0.9, and 100 samples. These aggressive parameters allow the model to explore a wider range of possibilities, useful when multiple completion attempts are possible. Due to the higher computational costs, we limited our evaluation to a subset of models compared to the main evaluation.

**Metrics**    For this evaluation, we expand our metrics to include *pass@10* and *pass@100*, alongside the standard *pass@1* and *ExcessCode*. *pass@10* and *pass@100* offer deeper insights into model performance by evaluating the success rate across the top 10 and 100 completions, respectively. These metrics are crucial for understanding how models perform in scenarios that permit multiple attempts, such as when users are provided with a range of completions to select from or when an external verifier is used to determine the best completion.

### A.2.1   Results

The results of our evaluation at higher sampling parameters are shown in Table 5. We draw several conclusions from the results.

***pass@1* decreases for open models.** Closed models maintain consistent *pass@1* performance under higher sampling parameters. In contrast, open source models generally exhibit a decline, showing a 2-3% reduction in *pass@1* performance compared to the main evaluation (Table 3).

**Multiple trials benefit open source models.** Open source models significantly improve with multiple trials, showing larger gains in *pass@10* and *pass@100* compared to closed models, which also improve but to a lesser degree. Specifically, EDITCODER-33b outperforms all models, including GPT-4, in *pass@100* for descriptive instructions and matches closely in *pass@10*. However, EDITCODER-33b lags behind GPT-4 in lazy instruction scenarios across all metrics and tends to generate more excess code for both prompt types. We expect that the performance of EDITCODER on lazy instructions will improve with more data and larger pre-trained models.

**Lazy instructions benefit more than descriptive instructions from multiple trials.** The performance disparity between descriptive and lazy instructions persists, even under higher sampling parameters and multiple trials, as seen in *pass@10* and *pass@100*. Despite this, the rate of improvement from multiple trials is greater for lazy instructions, with increases of 57.76% in *pass@10* and 22.57% in *pass@100*, surpassing the gains for descriptive instructions, which are 48.18% and 17.23% respectively. This indicates a more pronounced benefit from multiple attempts in scenarios involving less structured prompts.

**Significant increase in *ExcessCode*.** The average *ExcessCode* metrics for both descriptive and lazy instructions, at 0.507 and 0.449 respectively, have increased from the main evaluation's averages of 0.392 and 0.235. [1] This is expected, as higher sampling parameters tend to yield a broader range of completions, consequently resulting in an increase in superfluous code.

## A.3  Results Per Change Type

| Model | | Corrective | | Adaptive | | Perfective | |
|---|---|---|---|---|---|---|---|
| Name | Size | *p@1* | *ExcessCode* | *p@1* | *ExcessCode* | *p@1* | *ExcessCode* |
| GPT-4 | — | 62.21 | $0.05 \pm 0.03$ | 57.29 | $0.31 \pm 0.19$ | 53.43 | $0.08 \pm 0.06$ |
| GPT-3.5-Turbo | — | 47.93 | $0.00 \pm 0.00$ | 42.29 | $0.17 \pm 0.12$ | 46.07 | $0.60 \pm 0.54$ |
| EDITCODER | 33b | 56.86 | $0.02 \pm 0.02$ | 51.21 | $0.77 \pm 0.42$ | 39.29 | $0.05 \pm 0.04$ |
| EDITCODER | 6.7b | 48.64 | $0.00 \pm 0.00$ | 42.71 | $0.43 \pm 0.21$ | 40.07 | $0.66 \pm 0.42$ |
| EDITCODER | 1.3b | 26.36 | $0.11 \pm 0.10$ | 23.21 | $0.14 \pm 0.10$ | 22.57 | $0.26 \pm 0.18$ |

Table 6: Results of OpenAI models and EDITCODER on CANITEDIT per change type. We report the *pass@1* and *ExcessCode* metrics for each change type, as well as the size of the model if available. Results for lazy and descriptive prompts are aggregated across all problems. *pass@1* is abbreviated to *p@1*. Closed models are the rows highlighted in gray.

In this section we analyze the results of EDITCODER against the OpenAI models on CANITEDIT per change type. We hope to gain insights into the strengths and weaknesses of these models for different types of code changes. Table 6 shows the results of our analysis, we aggregate the results for lazy and descriptive prompts across all problems, this is done for conciseness and to minimize the noise of our results, as each *pass@1* and *ExcessCode* metric is calculated across 70 problems per change type, instead of 35. We utilize the same sampling parameters as in our main evaluation. Our key findings include:

- GPT-4 outperforms other models in functional correctness across all types of changes.

- Corrective changes typically incur minimal excess code, with some models achieving perfect scores in this area. Adaptive changes, intuitively, tend to introduce the most excess code.

---

[1]These values are calculated by averaging over the results from the models in this table, and not the entire set of models in the main evaluation.

- Our EDITCODER-33b model surpasses GPT-3.5-Turbo in both corrective and adaptive changes, while EDITCODER-6.7b shows comparable performance to GPT-3.5-Turbo in these categories. However, for perfective changes, both EDITCODER-33b and EDITCODER-6.7b underperform compared to GPT-3.5-Turbo. This suggests a potential improvement in training data for perfective changes. As demonstrated in Figure 3, verbs associated with perfective changes, such as *refactor* or *improve*, appear less frequently than those related to corrective or adaptive changes, such as *fix* or *add*, respectively. Artificially balancing the dataset with more examples of perfective changes could potentially enhance EDITCODER's performance. We leave this as an area for future work.

- According to our dataset, the most challenging changes are perfective, followed by adaptive, with corrective being the simplest.

## A.4  Evaluation on HumanEvalPack

| Model | | | *pass@1* | |
|---|---|---|---|---|
| Name | Size | Fix | Synthesize |
| GPT-4 | — | 47.0[‡] | **86.6**[‡] |
| EDITCODER | 33b | **53.0** | 63.5 |
| DeepSeekCoder-Instruct | 33b | 47.5[†] | 79.2 |
| CodeLlama-Instruct | 34b | 36.5[†] | 43.8 |
| StarCoder2 | 15b | 48.6[†] | 51.6 |
| StarCoderBase | 15b | 25.6[†] | 33.6[‡] |
| OctoCoder | 15b | 30.4[†] | 35.1[‡] |
| CodeLlama-Instruct | 13b | 19.4[†] | 23.7 |
| EDITCODER | 6.7b | 46.6 | 52.9 |
| DeepSeekCoder-Instruct | 6.7b | 44.9[†] | 76.2 |
| EDITCODER | 1.3b | 24.3 | 28.3 |
| DeepSeekCoder-Instruct | 1.3b | 9.1 | 62.4 |

Table 7: Results of models on the Python subset of HumanEvalFix and HumanEvalSynthesize. † indicates that the result originates from Lozhkov et al. (2024), while ‡ indicates that the result comes from Muennighoff et al. (2023). The rest of the results are from our evaluation following the same methodology as in Muennighoff et al. (2023).

To draw similarities and differences between CANITEDIT and HumanEvalPack, in this section we evaluate models on the Python subset of HumanEvalFix and HumanEvalSynthesize. Results are available in Table 7.

**Benchmark Overview**  HumanEvalPack is a benchmark comprised of 164 single-function problems aimed at evaluating both code generation and code editing. The problems are designed to not require domain-specific knowledge or familiarity with popular external libraries. HumanEvalFix contains only corrective code changes, while HumanEvalSynthesize purely focuses on code generation, tasking the model to generate a function from its signature and docstring.

**Results**  In this benchmark, EDITCODER-33b outperforms even GPT-4 in fixing bugs, while maintaining competitive synthesis capabilities against models trained on general instructional data, with EDITCODER-6.7b outperforming even larger models like CodeLlama-Instruct-34b. Additionally, we find a large disparity between the performance of models on HumanEvalFix and HumanEvalSynthesize for DeepSeekCoder-Instruct-1.3b, which performs significantly worse in fixing bugs than in synthesizing functions. These results demonstrate that HumanEvalPack and CANITEDIT complement each other: the former focuses on single-function algorithmic and puzzle-like problems, while the latter emphasizes

code editing tasks requiring broader knowledge of software engineering concepts in a wide range of domains.

**Example Problems**  To illustrate typical problems in HumanEvalPack, we selected examples from HumanEvalSynthesize and HumanEvalFix. Listing 2 presents a HumanEval-Synthesize problem where the model must complete a function based on its signature and docstring. Listing 3 demonstrates an incorrect function implementation from HumanEval-Fix along with it's ground truth unit test suite, where the model's task is to correct the implementation. Unlike in CANITEDIT, the model must infer the correct implementation solely from the faulty code and the test suite, without explicit instructions. One could argue that this falls under *intrinsic code editing*, rather than instructional code editing, since the model is not given any instructions about the intent of the function, making this benchmark more suitable for evaluating works such as Li et al. (2023a) and Gupta et al. (2023).

```
1  Write a Python function `has_close_elements(numbers: List[float],
2  threshold: float) -> bool` to solve the following problem:
3  Check if in given list of numbers, are any two numbers closer to
4  each other than given threshold.
5  >>> has_close_elements([1.0, 2.0, 3.0], 0.5)
6  False
7  >>> has_close_elements([1.0, 2.8, 3.0, 4.0, 5.0, 2.0], 0.3)
8  True
```

Listing 2: The prompt of a problem in HumanEvalSynthesize. The task for the model is to complete the function.

```
1  def unique_digits(x):
2      odd_digit_elements = []
3      for j, i in enumerate(x):
4          if all (int(c) % 2 == 1 for c in str(i)):
5              odd_digit_elements.append(i)
6              odd_digit_elements.append(j)
7      return sorted(odd_digit_elements)
8
9  def check(unique_digits):
10     assert unique_digits([15, 33, 1422, 1]) == [1, 15, 33]
11     assert unique_digits([152, 323, 1422, 10]) == []
12     assert unique_digits([12345, 2033, 111, 151]) == [111, 151]
13     assert unique_digits([135, 103, 31]) == [31, 135]
14
15  check(unique_digits)
16
17  Fix bugs in unique_digits.
```

Listing 3: The prompt of a problem in HumanEvalFix. The implementation is incorrect, and the task for the model is to re-implement the function correctly.

**Correlation with CANITEDIT**  We hypothesize that the performance of models on CAN-ITEDIT is correlated with their performance on general code generation tasks. Intuitively, if a model exhibits strong capabilities in code generation tasks, it is likely to perform well on code editing tasks as well. We quantify the correlation between the performance of models on CANITEDIT and HumanEvalPack by calculating a Pearson correlation coefficient between the *pass@1* metric of models on CANITEDIT and HumanEvalPack Synthesize, utilizing models on which we have results for both benchmarks. We find a correlation of 0.661, indicating a moderate positive correlation between the two benchmarks.

### A.5  Comparison of StarCoder Models

**StarCoder Models**  The first version of StarCoder models were pre-trained on several gigabytes of GitHub commits, as discussed in (Li et al., 2023b). In contrast, the second

| Model | | Descriptive | | Lazy | |
|---|---|---|---|---|---|
| Name | Size | *pass@1* | *ExcessCode* | *pass@1* | *ExcessCode* |
| StarCoder2 | 15b | **41.95** | $0.36 \pm 0.20$ | **31.48** | $0.04 \pm 0.04$ |
| StarCoder | 15b | 37.10 | $0.56 \pm 0.28$ | 27.62 | $0.42 \pm 0.34$ |
| StarCoderBase | 15b | 35.33 | $1.55 \pm 0.89$ | 27.05 | $0.85 \pm 0.55$ |
| StarCoderBase | 7b | 32.90 | $0.43 \pm 0.17$ | 21.95 | $0.49 \pm 0.37$ |
| StarCoder2 | 7b | 25.10 | $1.47 \pm 0.78$ | 13.76 | $1.81 \pm 1.22$ |
| StarCoder2 | 3b | 15.95 | $0.91 \pm 0.39$ | 13.33 | $1.09 \pm 0.98$ |
| StarCoderBase | 3b | 14.81 | $1.22 \pm 0.52$ | 9.90 | $1.17 \pm 0.76$ |
| StarCoderBase | 1b | 4.90 | $0.99 \pm 0.85$ | 5.48 | $0.00 \pm 0.00$ |

Table 8: Evaluation results of StarCoder models on CANITEDIT.

version, StarCoder2, did not include commit data in its training process. However, it was trained on GitHub issues, which provides it instruction following capabilities. Issue data encompasses a broader scope than commits, including discussions, bug reports, and feature requests. With prompt-engineering, StarCoder2 models can be used for code editing tasks, as demonstrated in Lozhkov et al. (2024). Furthermore, in most benchmarks evaluated in Lozhkov et al. (2024), StarCoder2 surpasses its previous version, StarCoderBase, in code generation tasks.

The original StarCoder model, StarCoderBase-15b, was additionally trained on Python code from GitHub. StarCoderBase models are available in four sizes: 15b, 7b, 3b, and 1b. On the other hand, StarCoder2 has not undergone further training on additional Python code and is available in three sizes: 15b, 7b, and 3b.

**Evaluation** We evaluate all sizes of StarCoder and StarCoder2 models on CANITEDIT and present the results in Table 8. We find that StarCoder2 outperforms StarCoder in the 15b and 3b sizes, but not in the 7b size, where StarCoderBase-7b significantly outperforms StarCoder2-7b. Additionally, for the 7b and 3b sizes, StarCoder2 models tend to generate more excess code than StarCoderBase models. We attribute the performance improvements in StarCoder2 to the broader training data and architectural enhancements, as discussed in Lozhkov et al. (2024), rather than to the superiority of issue data over commit data for code editing tasks. However, we also believe that for utilizing StarCoder2 models directly for code editing tasks, the issue prompt format is a viable alternative to the commit format previously utilized by StarCoderBase models. We provide the prompt format we utilized for StarCoder2 models in Figure 5.

## A.6 Prompt Templates Used in Evaluation

We evaluate all of our models on CANITEDIT using the same evaluation pipeline. However, for each model, we may utilize different prompts to generate the completions. These prompts are most aligned to how the model was trained, and are intended to maximize the model's performance on the task, while keeping the prompts as similar as possible across models. Figure 4 shows the prompts used for each model. For a fair comparison, we evaluate all models not trained on commits or explicit code editing tasks using a basic 1-shot prompt, showing the model how to add a sub function to a code segment with a add function, and changing the variable names from a and b to x and y.

Furthermore, given the natural language characteristics of GitHub issue data, significant prompt-engineering was required to facilitate code editing tasks for StarCoder2 models. The specific prompt format used for StarCoder2 models is provided separately in Figure 5.

```
<user>
You are PythonEditGPT. You will be
provided the original code snippet and
an instruction that specifies
the changes you need to make.
You will produce the changed code, based
on the original code and the instruction
given. Only produce the code, do not
include any additional prose.

## Code Before
```py
def add(a, b):
    return a + b
```

## Instruction
Add a "sub" function that subtracts two
numbers. Also write docstrings for both
functions and change a,b to x,y.
<assistant>
## Code After
```py
def add(x, y):
    """Adds two numbers."""
    return x + y
def sub(x, y):
    """Subtracts two numbers."""
    return x - y
```
<user>
You are PythonEditGPT. You will be
provided the original code snippet and
an instruction that specifies
the changes you need to make.
You will produce the changed code, based
on the original code and the instruction
given. Only produce the code, do not
include any additional prose.

## Code Before
```py
{before}
```

## Instruction
{instruction}
```

(a) Conversation template utilized for all chat models without a 'system' prompt. This is the prompt utilized for OctoCoder.

```
<system>
You are PythonEditGPT. You will be
provided the original code snippet and
an instruction that specifies
the changes you need to make.
You will produce the changed code, based
on the original code and the instruction
given. Only produce the code, do not
include any additional prose.
<user>
## Code Before
```py
def add(a, b):
    return a + b
```

## Instruction
Add a "sub" function that subtracts two
numbers. Also write docstrings for both
functions and change a,b to x,y.
<assistant>
## Code After
```py
def add(x, y):
    """Adds two numbers."""
    return x + y
def sub(x, y):
    """Subtracts two numbers."""
    return x - y
```
<user>
## Code Before
```py
{before}
```
## Instruction
{instruction}
```

(b) Conversation template utilized for all chat models with a 'system' prompt. The prompt is then adapted to the specific model chat format. This is the prompt utilized for: GPT-4, GPT-3.5-Turbo, CodeLlama-Instruct, and DeepSeekCoder-Instruct models.

```
<commit_before>
{before}
<commit_msg>
{instruction}
<commit_after>
```

(c) Prompt utilized for StarCoder and StarCoder-Base models of all sizes. StarCoder models are trained on commits in this format (Li et al., 2023b).

```
## Code Before:
{before}
## Instruction:
{instruction}
## Code After:
```

(d) Prompt utilized for our fine-tuned EDITCODER models. DeepSeekCoder-Base models use this prompt with the add and sub 1-shot example.

Figure 4: Prompts for each model evaluated on CANITEDIT. The {before} identifier is replaced with the 'before' code segment, and {instruction} is replaced with the instruction. Text wrapped in <...> is used to represent special tokens that utilized by the models.

```
<issue_start>username_0: I have a program in Python that I'd like to change.

Here is the code for the program:
```py
def add(a, b):
    return a + b
```

The change I'd like to make is:
Add a "sub" function that subtracts two numbers.
Also write docstrings for both functions and change a,b to x,y.

Please someone help me. Can you also provide the full code with the change?
<issue_comment>username_1: Sure, no problem. I will be able to help.
I am an expert in editing Python code.

Here is the full code with the change:
```py
def add(x, y):
    \"\"\"Adds two numbers.\"\"\"
    return x + y

    def sub(x, y):
    \"\"\"Subtracts two numbers.\"\"\"
    return x - y
```
Upvotes: 200<issue_comment>username_0: Thank you so much!
I have another program in Python that I'd like to change.

Here is the code for the program:
```py
{before}
```

The change I'd like to make is:
{instruction}

Please someone help me. Can you also provide the full code with the change?
Upvotes: 100<issue_comment>username_1: Sure, no problem. I will be able to help.
I am an expert in editing Python code.

Here is the full code with the change:
```py
{after}
```
```

Figure 5: Prompt utilized for StarCoder2 models. StarCoder2 models are trained on GitHub issue data, which makes this prompt format amenable to code editing tasks (Lozhkov et al., 2024).

## B  Training Details

In this section we provide details on the training process for EDITCODER and ablation results on the loss masking technique used in training.

### B.1  Training Tools and Configuration

For training all of our EDITCODER models, we utilize a fine-tuning pipeline based on the HuggingFace Transformers library (Wolf et al., 2020). Additionally, we utilize DeepSpeed ZeRO 3 (Rajbhandari et al., 2020) to efficiently shard the model across multiple GPUs. Due to memory constraints, we offload the optimizer to the CPU for the 33b model. We also use FlashAttention 2 (Dao, 2023) to speed up training on large context window sizes. All of our models are trained on a single machine equipped with 8 NVIDIA H100 (80GB) HGX GPUs. The effective micro-batch size is set at 32 (4 gradient accumulation steps, with a single batch per GPU). We utilize the AdamW optimizer with a learning rate of $2 \times 10^{-5}$, a linear decay scheduler, and 10 warmup steps. These parameters were chosen based on previous work on fine-tuning for code generation tasks (Cassano et al., 2023a), it is likely that we could get superior results by running a hyperparameter search. To facilitate reproducibility, we set the random seed to 42 for all experiments.

Prior to training, we shuffled the dataset randomly and deduplicated[2] it following the method outlined by Li et al. (2023b). This process combines MinHash (Broder, 2000) and Locality Sensitive Hashing (LSH) (Leskovec et al., 2014). We format the training data as a prompt, with the 'before' code segment followed by the 'instruction' and the 'after' code segment, and mask the loss calculation to only consider the 'after' code segment. All models underwent training for 8 epochs, with a packed context window of 8192 tokens, including padding for the remaining tokens. We select the model from the epoch with the highest performance on a held-out validation set. The number of epochs chosen for each EDITCODER is the following:

- EDITCODER-1.3b: 8
- EDITCODER-6.7b: 4
- EDITCODER-33b: 2

As shown by the number of epochs, we found that larger models overfit to the data more quickly, suggesting that we could achieve better results with a larger dataset.

### B.2  Effect of Loss Masking

In our training pipeline, we mask the loss calculation to only consider the 'after' code segment. The intuition behind this is that we don't need the model to learn how to reproduce the 'before' and 'instruction' segments, as these are always going to be provided as input to the model at inference time. The exact prompt format we use for training is shown in Figure 4d.

We wish to verify that this loss masking is beneficial for our task. To assess our hypothesis, we train two DeepSeekCoder-6.7b-Base models on EditPackFT[3], one with loss masking and one without, calculating the loss on all tokens. We then evaluate both models on CANITEDIT, and report the *pass@1* and *ExcessCode* metrics in Figure 6. The reported result with loss masking is the same as the one reported in Table 4. We find that the model trained with loss masking outperforms the model trained without it, and leads to a decrease in *ExcessCode* and its standard error. Furthermore, we plot the training loss curves for both models in Figure 7. We observe that the model trained with the loss masking technique is more stable and converges faster than the model trained without it.

---

[2]Deduplication, achieved by concatenating the 'before' and 'after' code segments, helps mitigate overfitting to specific training examples (Lee et al., 2022).

[3]We chose EditPackFT for this experiment as it is the smallest dataset we use for training, allowing us to quickly compare the two methods.

| Masking | pass@1 | ExcessCode |
|---------|--------|------------|
| Yes | **41.6** | $0.26 \pm 0.14$ |
| No | 40.5 | $0.43 \pm 0.20$ |

Figure 6: Effect of loss masking on the performance of DeepSeekCoder-6.7b-Base on EditPackFT.

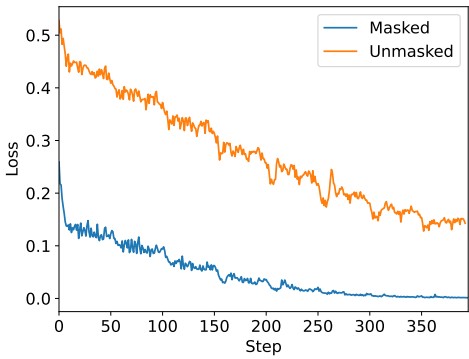

Figure 7: Training loss curves for DeepSeekCoder-6.7b-Base trained on EditPackFT with and without loss masking.

## C  Example CANITEDIT Benchmark Problems

We showcase four examples from the CANITEDIT benchmark, which we believe are representative of the types of problems present in the dataset.

**External Libraries in CANITEDIT**   Our benchmark includes 21 problems that import external libraries, which are libraries outside of Python's standard environment. We report the list of external libraries used and their number of appearances in the dataset: NumPy (13), Pandas (6), SciPy (3), scikit-learn (3), PyTorch (3), Z3 (2), autograd (2), Flask (1), vLLM (1)

**oop_refactor**   Figure 8 details a task where the model refactors code using object-oriented programming (OOP) principles. Initially, the code is a function for formatting messages based on type. The refactoring involves creating `TextMessage` and `ImageMessage` as subclasses of an abstract `Message` class and implementing a `MessageFactory` for message construction. This task provides an example of a *perfective* edit, focusing on reorganizing the code into an OOP style without adding new features. The transformation is quite significant, and the largest relative transformation in our dataset: from a single function to a multi-class OOP program. The goal is to assess the model's proficiency in converting functional code into well-structured OOP designs based on comprehensive instructions and for the model to restructure small programs into much larger ones. Our test suites verify both functional correctness and the proper hierarchical class structure.

**group_theory**   Figure 9 features a task to modify a class from representing group $C4$ to group $C8$, including its operations like inverse and product. The problem highlights domain-specific problems in CANITEDIT, this one being set in the context of cyclic groups. Testing domain-specific edits is crucial, especially when comparing the capabilities of large proprietary models like GPT-4 with smaller open models. It requires the model to transform the `C4` class (representing a 4-element cyclic group) into the `C8` class (for an 8-element group), requiring extensive edits across various code sections. This complexity presents a significant test for other code editing approaches, such as fill-in-the-middle (Bavarian et al., 2022; Fried et al., 2023), which may struggle with multiple edit locations (Yee & Guha, 2023). Key edits involve altering the `size` and `elements` methods. The necessary understanding for these modifications stems from group theory, which is not explicitly explained in the problem. This setup tests the model's capability to execute domain-specific edits where contextual knowledge is implied rather than provided.

**strategy**   Figure 10 presents an open-ended problem where the model devises a game strategy to defeat the already implemented `CornerStrategy` in Tic Tac Toe. This task represents an *adaptive* edit, focused on developing a new feature without altering existing classes. The uniqueness in this problem lies in the lack of providing rules for the game, but rather requiring the model to infer them through understanding of the code. Additionally, it leaves the strategy design entirely to the model's discretion. Our tests ensure that the `Game` class remain intact and that the model's strategy consistently outperforms `CornerStrategy` in the game.

**sudoku_solver**   Figure 11 presents a sudoku solver problem leveraging the Z3 satisfiability modulo (SMT) solver. The problem starts with an incomplete solver that lacks checks for 3x3 subgrids, both in its solving logic and board validity function. In sudoku, each 3x3 grid must contain distinct numbers from 1 to 9. The task involves adding these checks to ensure the solver can correctly solve a sudoku board. This problem assesses the model's capability to implement edits across different code sections. Although it uses Z3, in-depth knowledge of the library or SMT isn't required; the necessary features needed to solve the problem can be inferred from the existing code, which already includes checks for row and column uniqueness.

```
1  def process_message(message, message_type):
2      if message_type == "text":
3          return f"Processed text message: {message}"
4      elif message_type == "image":
5          return f"Processed image message with description: {message}"
6      else:
7          return "Unknown message type"
8
```

(a) 'before' code segment of the oop_refactor problem (Figure 8).

| Instruction | Type |
|---|---|
| Abstract the code into an object-oriented version of itself. To do that, create an abstract class `Message(ABC)`, which can be initialized with a `content` string. The class should have an abstract method `process(self)`, which should return a string. Create two children classes `TextMessage` and `ImageMessage`, which implement the `process` method. Finally, create a `MessageFactory` that has a static method `get_message(message_type, content) -> Message`; static methods can be defined with the `@staticmethod` decorator. The `get_message` method should return `Message` corresponding to the `message_type` (either `text` or `image`), and it should throw a ValueError if the `message_type` is not valid. | **Descriptive** |
| Make the code object-oriented. Specifically, create an abstract class `Message`, and children classes `TextMessage` and `ImageMessage`. The `Message` class should have a method `process(self)` that returns the message which was given to the constructor. Also, create a `MessageFactory` that has a static method `get_message(message_type, content) -> Message`; should raise an exception if the message type is not supported. | **Lazy** |

(b) Instructions for the oop_refactor problem (Figure 8).

```
1  from abc import ABC, abstractmethod
2
3  class Message(ABC):
4      """
5      Abstract class for messages
6      """
7      def __init__(self, content):
8          self.content = content
9
10     @abstractmethod
11     def process(self):
12         pass
13
14 class TextMessage(Message):
15     """
16     Concrete class for TextMessage
17     """
18     def process(self):
19         return f"Processed text message: {self.content}"
20
21 class ImageMessage(Message):
22     """
23     Concrete class for ImageMessage
24     """
25     def process(self):
26         return f"Processed image message with description: {self.content}
       "
27
28 class MessageFactory:
29     """
30     Factory class for creating message objects
31     """
32     @staticmethod
33     def get_message(message_type, content):
34         if message_type == "text":
35             return TextMessage(content)
36         elif message_type == "image":
37             return ImageMessage(content)
38         else:
39             raise ValueError("Unknown message type")
40
```

(c) Canonical solution for the oop_refactor problem (Figure 8).

Figure 8: The oop_refactor problem from CANITEDIT. This is a prime example of a **perfective** type of edit, as asks the model to refactor code using OOP principles, without adding any additional features.

```
1  import torch
2  import numpy as np
3  import torch.nn as nn
4
5  class C4(nn.Module):
6      """Represents the C4 class of group theory, where each
7         element represents a discrete rotation."""
8
9      def __init__(self):
10         super().__init__()
11         self.register_buffer('identity', torch.Tensor([0.]))
12
13     def size(self):
14         """Outputs the size of this group."""
15         return 4
16
17     def elements(self):
18         """Returns all the elements of this group"""
19         return torch.tensor([0., np.pi / 2, np.pi, 3 * np.pi / 2])
20
21     def product(self, h, g):
22         """Compute the product of two elements g and h in the group C4"""
23         return torch.remainder(h + g, 2 * np.pi)
24
25     def inverse(self, h):
26         """Computes the inverse of the element h in the group C4"""
27         return torch.remainder(-h, 2 * np.pi)
28
29     def matrix_representation(self, h):
30         """Returns the matrix representation of this element"""
31         cos_t = torch.cos(h)
32         sin_t = torch.sin(h)
33         representation = torch.tensor([
34             [cos_t, -sin_t],
35             [sin_t, cos_t]
36         ], device=self.identity.device)
37         return representation
38
```

(a) 'before' code segment of the group_theory problem (Figure 9)

| Instruction | Type |
|---|---|
| Edit the C4 class, which represents rotations of 0, 90, 180 and 270 degrees, to represent the class C8, which represents rotations of 0, 45, 90, 135, 180, 225, 270 and 315 degrees. | **Descriptive** |
| Edit the C4 class and its methods to represent the C8 group instead | **Lazy** |

(b) Instructions for the group_theory problem (Figure 9).

```python
1   import torch
2   import numpy as np
3   import torch.nn as nn
4
5   class C8(nn.Module):
6       """Represents the C8 class of group theory, where each
7       element represents a discrete rotation."""
8
9       def __init__(self):
10          super().__init__()
11          self.register_buffer('identity', torch.Tensor([0.]))
12
13      def size(self):
14          """Outputs the size of this group."""
15          return 8
16
17      def elements(self):
18          """Returns all the elements of this group"""
19          delta = np.pi / 4
20          return torch.tensor([0., delta, delta * 2, delta * 3,
21                               delta * 4, delta * 5, delta * 6, delta * 7])
22
23      def product(self, h, g):
24          """Compute the product of two elements g and h in the group C8"""
25          return torch.remainder(h + g, 2 * np.pi)
26
27      def inverse(self, h):
28          """Computes the inverse of the element h in the group C8"""
29          return torch.remainder(-h, 2 * np.pi)
30
31      def matrix_representation(self, h):
32          """Returns the matrix representation of this element"""
33          cos_t = torch.cos(h)
34          sin_t = torch.sin(h)
35          representation = torch.tensor([
36              [cos_t, -sin_t],
37              [sin_t, cos_t]
38          ], device=self.identity.device)
39          return representation
40
```

(c) Canonical solution for the group_theory problem (Figure 9).

Figure 9: The group_theory problem from CANITEDIT. This exemplifies the subset of domain-specific problems in our benchmark.

```
1   from abc import ABC
2   from abc import abstractmethod
3   from typing import List, Tuple
4
5   class Strategy(ABC):
6       @abstractmethod
7       def returnMove(self, board: List[List[bool]]) -> Tuple[int, int]:
8           '''Returns a tuple(row, column) which indicates where to move
9             in a 3x3 grid.'''
10          pass
11
12  class CornerStrategy(Strategy):
13      def returnMove(self, board: List[List[bool]]) -> Tuple[int, int]:
14          if board[0][0] == None: return (0, 0)
15          elif board[0][2] == None: return (0, 2)
16          elif board[2][0] == None: return (2, 0)
17          elif board[2][2] == None: return (2, 2)
18          else: raise Exception
19
20  class Game:
21      def __init__(self, player1: Strategy, player2: Strategy):
22          self.playerOne = player1
23          self.playerTwo = player2
24          self.board = [[None for _ in range(3)] for _ in range(3)]
25
26      def player1Won(self):
27          playerTurn = True
28          while (not self.playerXWon(True) and not self.playerXWon(False)
29          and not self.gameOver()):
30              strat = self.playerOne if playerTurn else self.playerTwo
31              move = strat.returnMove(self.board)
32              self.board[move[0]][move[1]] = playerTurn
33              playerTurn = not playerTurn
34          if self.gameOver(): return False
35          else: return self.playerXWon(True)
36
37      def gameOver(self):
38          for row in self.board:
39              for col in row:
40                  if col == None: return False
41          return True
42
43      def playerXWon(self, x: bool):
44          for i in range(3):
45              if self.rowNX(i, x): return True
46          for i in range(3):
47              if self.colNX(i, x): return True
48          downDiag = self.board[0][0] == x and self.board[1][1] == x and
          self.board[2][2] == x
49          upDiag = self.board[2][0] == x and self.board[1][1] == x and self
          .board[0][2] == x
50          return downDiag or upDiag
51
52      def rowNX(self, n: int, x: bool):
53          for col in self.board[n]:
54              if col != x: return False
55          return True
56
57      def colNX(self, n: int, x: bool):
58          for row in self.board:
59              if row[n] != x: return False
60          return True
61
```

(a) 'before' code segment of the strategy problem (Figure 10).

| Instruction | Type |
|---|---|
| The following code describes a tic-tac-toe game which takes in two strategies and determines who wins if they play each other. The `Strategy` class defines an abstract method, `returnMove(board)`, which returns a tuple representing where this strategy will move, given a board state. The `CornerStrategy` class is a subclass of `Strategy` with a concrete implementation of `returnMove(board)`. The `Game` class constructor takes in two strategies. It has a method `player1Won` which determines if the first strategy provided will beat the other if they both take turns alternating between moves. There are two methods, `playerXWon` and `gameOver` which determine how a game is won and when it is over. Create a class `GoodStrategy` which extends `Strategy` such that `Game(GoodStrategy(), CornerStrategy()).player1Won()` returns `True`. This can not be solved by modifying the `Game`, `Strategy`, or `CornerStrategy` classes in any way. | Descriptive |
| Create a strategy `GoodStrategy`, that beats `CornerStrategy`. Do not modify the `Game` class. | Lazy |

(b) Instructions for the strategy problem (Figure 10).

```python
1   from abc import ABC
2   from abc import abstractmethod
3   from typing import List, Tuple
4
5   class Strategy(ABC):
6       @abstractmethod
7       def returnMove(self, board: List[List[bool]]) -> Tuple[int, int]:
8           '''Returns a tuple(row, column) which indicates where to move
9              in a 3x3 grid.'''
10          pass
11
12  class CornerStrategy(Strategy):
13      def returnMove(self, board: List[List[bool]]) -> Tuple[int, int]:
14          if board[0][0] == None: return (0, 0)
15          elif board[0][2] == None: return (0, 2)
16          elif board[2][0] == None: return (2, 0)
17          elif board[2][2] == None: return (2, 2)
18          else: raise Exception
19
20  class GoodStrategy(Strategy):
21      def __init__(self) -> None:
22          super().__init__()
23          self.turn = 0
24      def returnMove(self, board: List[List[bool]]) -> Tuple[int, int]:
25          self.turn += 1
26          if self.turn == 1: return (0, 1)
27          elif self.turn == 2: return (1, 1)
28          elif self.turn == 3: return (2, 1)
29          raise Exception
30
31  class Game:
32      def __init__(self, player1: Strategy, player2: Strategy):
33          self.playerOne = player1
34          self.playerTwo = player2
35          self.board = [[None for _ in range(3)] for _ in range(3)]
36      def player1Won(self):
37          ...
38      def gameOver(self):
39          ...
40      def playerXWon(self, x: bool):
41          ...
42      def rowNX(self, n: int, x: bool):
43          ...
44      def colNX(self, n: int, x: bool):
45          ...
46
```

(c) Canonical solution for the `strategy` problem (Figure 10).

Figure 10: The `strategy` problem from CANITEDIT. This problem is a prime example of a **adaptive** type of edit, and is characteristic in the open-endedness of the instructions, both descriptive and lazy.

```python
1   from typing import List, Optional
2   from z3 import ArithRef, Int, Solver, Distinct, And, sat, IntVal
3
4   def make_9x9_z3_board(board_text: str, solver: Solver) -> List[List[ArithRef]]:
5       """
6       Creates a board of z3 variables from a string representation of a board.
7       For unknown cells, make the value be 0, and for known cells, make the value
8       be a number from 1-9.
9       """
10      board = []
11      for line_counter, line in enumerate(board_text.splitlines()):
12          row = []
13          for char_counter, character in enumerate(line.strip()):
14              if character.isdigit():
15                  num = int(character)
16                  # 0 is unknown
17                  cell = Int(f"cell_{line_counter}_{char_counter}")
18                  if num == 0:
19                      solver.add(And(cell >= 1, cell <= 9))
20                      row.append(cell)
21                  elif 0 < num < 10:
22                      solver.add(cell == IntVal(num))
23                      row.append(cell)
24          if len(row) != 9:
25              raise ValueError(
26                  f"Invalid column count of board, must be 9, got {len(row)}")
27          board.append(row)
28
29      if len(board) != 9:
30          raise ValueError(
31              f"Invalid row count of board, must be 9, got {len(board)}")
32
33      return board
34
35  def assert_uniq(solver: Solver, z3_board: List[List[ArithRef]]):
36      # Assert rows unique
37      for row in z3_board:
38          solver.add(Distinct(row))
39
40      # Assert columns unique
41      for col in zip(*z3_board):
42          solver.add(Distinct(col))
43
44  def print_board(board: List[List[int]]):
45      for row in board:
46          print(row)
47
48  def check_valid(board: List[List[int]]) -> bool:
49      for row in board:
50          if len(set(row)) != 9:
51              return False
52
53      for col in zip(*board):
54          if len(set(col)) != 9:
55              return False
56
57      return True
58
59  def solve(board_text: str) -> Optional[List[List[int]]]:
60      solver = Solver()
61      z3_board = make_9x9_z3_board(board_text, solver)
62      board: List[List[int]] = [[] for _ in range(9)]
63      assert_uniq(solver, z3_board)
64      if solver.check() == sat:
65          model = solver.model()
66          for i, row in enumerate(z3_board):
67              row = [model.evaluate(cell).as_long()   # type: ignore
68                      for cell in row]
69              board[i] = row
70          return board
71      else: return None
72
```

(a) 'before' code segment of the sudoku_solver problem (Figure 11).

| Instruction | Type |
|---|---|
| This version of the sudoku solver and checker does not reflect the original game of sudoku; the original game also checks for the uniqueness of 3x3 subgrids in addition to the rows and columns. Update the `assert_uniq` function to add new constraints for all nine 3x3 subgrids, and update the `check_valid` function to make sure that input grids have unique 3x3 subgrids. | **Descriptive** |
| Make both the sudoku solver and verifier support the nine 3x3 subgrids that are in the original sudoku game. | **Lazy** |

(b) Instructions for the sudoku_solver problem (Figure 11).

```python
1   from typing import List, Optional
2   from z3 import ArithRef, Int, Solver, Distinct, And, sat, IntVal
3
4   def make_9x9_z3_board(board_text: str, solver: Solver) -> List[List[ArithRef]]:
5       ...
6
7   def assert_uniq(solver: Solver, z3_board: List[List[ArithRef]]):
8       # Assert rows unique
9       for row in z3_board:
10          solver.add(Distinct(row))
11
12      # Assert columns unique
13      for col in zip(*z3_board):
14          solver.add(Distinct(col))
15
16      # Assert 3x3 squares unique
17      for i in range(0, 9, 3):
18          for j in range(0, 9, 3):
19              square = [z3_board[x][y]
20                        for x in range(i, i+3) for y in range(j, j+3)]
21              solver.add(Distinct(square))
22
23  def print_board(board: List[List[int]]):
24      for row in board:
25          print(row)
26
27  def check_valid(board: List[List[int]]) -> bool:
28      for row in board:
29          if len(set(row)) != 9: return False
30
31      for col in zip(*board):
32          if len(set(col)) != 9: return False
33
34      for i in range(0, 9, 3):
35          for j in range(0, 9, 3):
36              square = [board[x][y]
37                        for x in range(i, i+3) for y in range(j, j+3)]
38              if len(set(square)) != 9: return False
39      return True
40
41  def solve(board_text: str) -> Optional[List[List[int]]]:
42      solver = Solver()
43      z3_board = make_9x9_z3_board(board_text, solver)
44      board: List[List[int]] = [[] for _ in range(9)]
45      assert_uniq(solver, z3_board)
46      if solver.check() == sat:
47          model = solver.model()
48          for i, row in enumerate(z3_board):
49              row = [model.evaluate(cell).as_long()  # type: ignore
50                     for cell in row]
51              board[i] = row
52          return board
53      else: return None
```

(c) Canonical solution for the sudoku_solver problem (Figure 11).

Figure 11: The sudoku_solver problem from CANITEDIT. This problem uses the Z3 theorem proving library, and is an example of a **corrective** type of edit, as it requires the model to correct an existing solver to include checks for 3x3 subgrids.

# D   Example Model Completions

This section analyzes various completions from the models we evaluated, displaying both correct and incorrect examples to highlight their strengths and weaknesses.

## D.1   Excess Code Generation

Figure 12 provides an instance of EDITCODER-1.3b generating excess code. This case underscores the importance of the ExcessCode metric ( §5.1), which penalizes models for generating unneeded code. Here, the model, while correctly altering the `flatten` method in `Tensor`, unnecessarily adds an `unflatten` method, resulting in an ExcessCode score of 66.67. To accurately measure ExcessCode, completions must pass the test suite for proper line coverage evaluation. Figure 13 shows another excess code example where the completion is failing the test suite, preventing the evaluation of the ExcessCode metric.

## D.2   Model-Specific Completion Capabilities

Different models exhibit unique competencies in completing different tasks. Figure 14 highlights a solution by EDITCODER-33b for a complex problem involving the modification of a maze solver class to use the A* algorithm with Manhattan distance heuristic instead of the existing Uniform Cost Search (UCS) algorithm. A common error observed, particularly in GPT-4, is the incorrect persistence of heuristic values in the A* algorithm, where the heuristic value persists across multiple iterations of the algorithm inside the frontier queue. In contrast, Figure 15 presents a problem solved exclusively by OpenAI models, requiring domain-specific knowledge on monte carlo approximation for kl-divergence, a skill less common in smaller open-source models. This illustrates the balance between using OpenAI models for domain-specific issues and open-source models for more general code editing tasks.

```
1  class Tensor:
2      def __init__(self, matrix):
3          self.matrix = matrix
4
5      def m(self):
6          return len(self.matrix)
7
8      def n(self):
9        return len(self.matrix[0])
10
11      def relu(self):
12          for i in range(self.m()):
13              for j in range(self.n()):
14                  self.matrix[i][j] = max(0, self.matrix[i][j])
15
16      def flatten(self):
17          sofar = []
18          for i in range(self.n()):
19              for j in range(self.m()):
20                  sofar.append(self.matrix[j][i])
21          return sofar
```

(a) 'before' code segment of Figure 12.

| **Edit Instruction** |
| --- |
| Change `flatten` to flatten lists left to right, top down. |

(b) Instruction for Figure 12.

```
1   class Tensor:
2       def __init__(self, matrix):
3           self.matrix = matrix
4
5       def m(self):
6           return len(self.matrix)
7
8       def n(self):
9         return len(self.matrix[0])
10
11      def relu(self):
12          for i in range(self.m()):
13              for j in range(self.n()):
14                  self.matrix[i][j] = max(0, self.matrix[i][j])
15
16      def flatten(self):
17          sofar = []
18          for i in range(self.m()):
19              for j in range(self.n()):
20                  sofar.append(self.matrix[i][j])
21          return sofar
22
23      def unflatten(self, flat):
24          self.matrix = []
25          for i in range(self.n()):
26              self.matrix.append([])
27              for j in range(self.m()):
28                  self.matrix[i].append(flat[i*self.n() + j])
```

(c) Model-generated completion for Figure 12.

Figure 12: An example of EDITCODER-1.3b generating excess code. Tasked with modifying the flatten method in Tensor, the model correctly alters flatten but also adds an unwanted unflatten method. This instance scores 66.67 on the ExcessCode metric.

```
1   import math
2   from collections import Counter
3   from typing import List, Dict
4
5   class BM25:
6       def __init__(
7           self,
8           corpus: List[List[str]],
9           k1: float = 1.5,
10          b: float = 0.75
11      ) -> None:
12          self.corpus_size = len(corpus)
13          self.doc_freqs = []
14          self.corpus = corpus
15          self.df = {}
16          self.idf = {}
17          self.doc_len = []
18          self.avgdl = -1
19          self._initialize()
20          self.k1 = k1
21          self.b = b
22
23      def _initialize(self) -> None:
24          for document in self.corpus:
25              frequencies = {}
26              self.doc_len.append(len(document))
27              for word in document:
28                  frequencies[word] = frequencies.get(word, 0) + 1
29              self.doc_freqs.append(frequencies)
30          for document in self.doc_freqs:
31              for word, freq in document.items():
32                  self.df[word] = self.df.get(word, 0) + 1
33          for word, freq in self.df.items():
34              self.idf[word] = math.log(1 + (self.corpus_size
35                  - freq + 0.5) / (freq + 0.5))
36          self.avgdl = sum(self.doc_len) / self.corpus_size
37
38      def calculate_bm25(
39          self,
40          document_index: int,
41          query: List[str]
42      ) -> float:
43          score = 0.0
44          document = self.corpus[document_index]
45          doc_len = len(document)
46          for term in query:
47              if term in self.doc_freqs[document_index]:
48                  term_freq = self.doc_freqs[document_index][term]
49                  score += self.idf[term] * term_freq
50                      * (self.k1 + 1)
51                      / (term_freq + self.k1 * (1 - self.b
52                          + self.b * doc_len / self.avgdl))
53          return score
54
55      def rank(self, query: List[str]) -> List[float]:
56          scores = [self.calculate_bm25(idx, query)
57              for idx in range(self.corpus_size)]
58          return scores
```

(a) 'before' code segment of Figure 13.

**Edit Instruction**

```
Optimize the bm25 algorithm by avoiding frequency calculations.
```

(b) Instruction for Figure 13

```python
1   import math
2   from typing import List, Dict
3
4   class BM25:
5       def __init__(
6           self,
7           corpus: List[List[str]],
8           k1: float = 1.5,
9           b: float = 0.75
10      ) -> None:
11          self.corpus = corpus
12          self.corpus_size = len(corpus)
13          self.avgdl = sum(len(doc) for doc in corpus)
14              / self.corpus_size
15          self.k1 = k1
16          self.b = b
17
18      def calculate_bm25(
19          self,
20          document_index: int,
21          query: List[str]
22      ) -> float:
23          doc_freqs: List[Dict[str, int]] = []
24          df: Dict[str, int] = {}
25          idf = {}
26          for document in self.corpus:
27              frequencies: Dict[str, int] = {}
28              for word in document:
29                  frequencies[word] = frequencies.get(word, 0) + 1
30                  if word not in df:
31                      df[word] = 0
32                  df[word] += 1
33              doc_freqs.append(frequencies)
34          for word, freq in df.items():
35              idf[word] = math.log(1 + (self.corpus_size
36                  - freq + 0.5) / (freq + 0.5))
37          score = 0.0
38          document = self.corpus[document_index]
39          doc_len = len(document)
40          for term in query:
41              if term in doc_freqs[document_index]:
42                  term_freq = doc_freqs[document_index][term]
43                  score += idf[term] * term_freq
44                      * (self.k1 + 1)
45                      / (term_freq + self.k1 * (1 - self.b
46                          + self.b * doc_len / self.avgdl))
47          return score
48
49      def rank(self, query: List[str]) -> List[float]:
50          scores = [self.calculate_bm25(idx, query)
51              for idx in range(self.corpus_size)]
52          return scores
```

(c) Model-generated completion for Figure 13.

Figure 13: An example of excess added code to the solution, which is incorrect - produced by the EDITCODER-33b. The model adds a "doc_len" instance attribute which is not necessary for the solution and is not implemented in the proposed solution shown above. In addition, it is not necessary to store the document lengths as it is a quick lookup to get the length of a list in Python.

```
1    from typing import List, Literal, Tuple
2    from queue import PriorityQueue
3
4    Move = Literal["up", "down", "left", "right"]
5    # 0 = up, 1 = down, 2 = left, 3 = right
6    MoveIndex = Literal[0, 1, 2, 3]
7    # 0 = empty, 1 = wall, 2 = start, 3 = end
8    Cell = Literal[0, 1, 2, 3]
9
10   class Maze:
11       def __init__(self, maze: List[List[Cell]]):
12           self.maze = maze
13           self.rows = len(maze)
14           self.cols = len(maze[0])
15           self.start = self.find_start()
16           self.end = self.find_end()
17
18       def find_start(self) -> Tuple[int, int]:
19           for row in range(self.rows):
20               for col in range(self.cols):
21                   if self.maze[row][col] == 2:
22                       return row, col
23           raise ValueError("No start found")
24
25       def find_end(self) -> Tuple[int, int]:
26           for row in range(self.rows):
27               for col in range(self.cols):
28                   if self.maze[row][col] == 3:
29                       return row, col
30           raise ValueError("No end found")
31
32       def get_neighbors(
33           self, row: int, col: int
34       ) -> List[Tuple[int, int]]:
35           neighbors = []
36           if row > 0 and self.maze[row - 1][col] != 1:
37               neighbors.append((row - 1, col))
38           if row < self.rows - 1 and self.maze[row + 1][col] != 1:
39               neighbors.append((row + 1, col))
40           if col > 0 and self.maze[row][col - 1] != 1:
41               neighbors.append((row, col - 1))
42           if col < self.cols - 1 and self.maze[row][col + 1] != 1:
43               neighbors.append((row, col + 1))
44           return neighbors
45
46       def solve(self) -> Tuple[int, List[Tuple[int, int]]]:
47           """Uses UCS to find a path from start to end, returning
48           the number of nodes expanded and the path if one
49           exists. The cost of each move is 1."""
50           visited = set()
51           frontier = PriorityQueue()
52           frontier.put((0, self.start, []))
53           expanded = 0
54           while not frontier.empty():
55               cost, current, path = frontier.get()
56               if current in visited:
57                   continue
58               visited.add(current)
59               new_path = path + [current]
60               if current == self.end:
61                   return expanded, new_path
62               for neighbor in self.get_neighbors(*current):
63                   if neighbor not in visited:
64                       new_cost = cost + 1
65                       frontier.put(
66                           (new_cost, neighbor, new_path))
67               expanded += 1
68           return expanded, []
```

(a) 'before' code segment of Figure 14.

**Edit Instruction**

```
Change the `solve` function to use A* with manhattan distance instead
of using UCS.
```

(b) Instruction for Figure 14.

```python
1   from typing import List, Literal, Tuple
2   from queue import PriorityQueue
3
4   Move = Literal["up", "down", "left", "right"]
5   # 0 = up, 1 = down, 2 = left, 3 = right
6   MoveIndex = Literal[0, 1, 2, 3]
7   # 0 = empty, 1 = wall, 2 = start, 3 = end
8   Cell = Literal[0, 1, 2, 3]
9
10  def manhattan_distance(
11      a: Tuple[int, int], b: Tuple[int, int]) -> int:
12      return abs(a[0] - b[0]) + abs(a[1] - b[1])
13
14  class Maze:
15      def __init__(self, maze: List[List[Cell]]):
16          self.maze = maze
17          self.rows = len(maze)
18          self.cols = len(maze[0])
19          self.start = self.find_start()
20          self.end = self.find_end()
21
22      def find_start(self) -> Tuple[int, int]:
23          for row in range(self.rows):
24              for col in range(self.cols):
25                  if self.maze[row][col] == 2:
26                      return row, col
27          raise ValueError("No start found")
28
29      def find_end(self) -> Tuple[int, int]:
30          for row in range(self.rows):
31              for col in range(self.cols):
32                  if self.maze[row][col] == 3:
33                      return row, col
34          raise ValueError("No end found")
35
36      def get_neighbors(
37          self, row: int, col: int
38      ) -> List[Tuple[int, int]]:
39          neighbors = []
40          if row > 0 and self.maze[row - 1][col] != 1:
41              neighbors.append((row - 1, col))
42          if row < self.rows - 1 and self.maze[row + 1][col] != 1:
43              neighbors.append((row + 1, col))
44          if col > 0 and self.maze[row][col - 1] != 1:
45              neighbors.append((row, col - 1))
46          if col < self.cols - 1 and self.maze[row][col + 1] != 1:
47              neighbors.append((row, col + 1))
48          return neighbors
49
50      def solve(self) -> Tuple[int, List[Tuple[int, int]]]:
51          """Uses A* with the manhattan distance as a heuristic
52          to find a path from start to end, returning the
53          number of nodes expanded and the path if one exists."""
54          visited = set()
55          # Each entry in the PQ is a tuple of
56          # (f(n), g(n), coord, path)
57          frontier = PriorityQueue()
58          frontier.put((0, 0, self.start, []))
59          expanded = 0
60          while not frontier.empty():
61              _, cost, current, path = frontier.get()
62              if current in visited: continue
63              visited.add(current)
64              new_path = path + [current]
65              if current == self.end:
66                  return expanded, new_path
67              for neighbor in self.get_neighbors(*current):
68                  if neighbor not in visited:
69                      g = cost + 1
70                      h = manhattan_distance(neighbor, self.end)
71                      f = g + h
72                      frontier.put((f, g, neighbor, new_path))
73              expanded += 1
74          return expanded, []
```

(c) Model-generated completion for Figure 14.

Figure 14: A correct code edit solution generated by EDITCODER-33b. The model correctly converts the UCS implementation to A* and adds a new manhattan_distance function as a standalone utility. Out of the models we evaluated, only EDITCODER-33b was able to solve this problem.

```
1  import torch
2
3  def kl_div(
4      q: torch.distributions.Distribution,
5      p: torch.distributions.Distribution
6  ) -> torch.Tensor:
7      return torch.distributions.kl_divergence(q, p).mean()
```

(a) 'before' code segment of Figure 15.

**Edit Instruction**

```
Change kl_div to compute a monte carlo approximation of the kl
divergence given num_samples as a parameter, which by default is set
to 100000.
```

(b) Instruction for Figure 15.

```
1  import torch
2
3  def kl_div(
4      q: torch.distributions.Distribution,
5      p: torch.distributions.Distribution,
6      num_samples: int = 100000
7  ) -> torch.Tensor:
8      samples = q.sample((num_samples,))
9      return (q.log_prob(samples) - p.log_prob(samples)).mean()
```

(c) Model-generated completion for Figure 15.

Figure 15: A correct code edit solution generated by GPT-4. Interestingly, GPT-4 can solve this problem while all fine-tuned models are unable to solve it. This problem requires knowledge of a sampling technique for approximating the KL divergence between two distributions.

# E   Using LLMs in Code Editing Tasks

In this section, we provide a brief overview of the use of LLMs in code editing tasks. We showcase two scenarios: (1) humans interacting with chat models to edit code, and (2) models automatically generating edits for code. For the former, we analyze a large dataset of LLM chatbot interactions, "lmsys/lmsys-chat-1m" which can be found on HuggingFace's hub, and for the latter, we analyze a sample reflection generated by GPT-4 using the Reflexion algorithm (Shinn et al., 2023).

## E.1   Human-Instructed Code Editing

```
1  def preprocess(original_document):
2      rows = build_rows(original_document)
3      packed_row_boxes = list(rows.values())
4
5      if len(packed_row_boxes) == 1:
6          return None, None
7
8      paragraph_classes = get_paragraphs(packed_row_boxes,
9                                         cluster_threshold=
       CLUSTER_THRESHOLD,
10                                        distance_threshold=
       DISTANCE_THRESHOLD)
11     paragraph_boxes = make_paragraph_from_prediction(packed_row_boxes,
       paragraph_classes)
12     # rest of code omitted for brevity
13
```

(a) **Instruction:**  Can you refactor this python code?

```
1  (function() {
2    var x = 10;
3    var y = 20;
4    var z = 30;
5    var a = function(b, c) {
6      return b + c;
7    };
8    var b = function(d, e) {
9      return d - e;
10   };
11   var c = function(f, g) {
12     return f * g;
13   };
14   console.log(a(x, y));
15   console.log(b(z, x));
16   console.log(c(x, y));
17 })();
```

(b) **Instruction:**  cool, now please refactor the snippet to have exactly the same logic and be as readable as possible.

Figure 16: Two example human editing requests taken from the "lmsys/lmsys-chat-1m" dataset which contains 1-million in-the-wild conversations from 25 conversational LLMs

We analyze a large dataset of human interactions with 25 different conversational LLMs, users to interact with a highly capable chatbot. The dataset, "lmsys/lmsys-chat-1m", contains 1-million real conversations from 25 conversational LLMs of varying sizes and capabilities. We analyze the dataset to understand how humans interact with LLMs to edit code. We find that 4188 of the 1-million conversations contain a code-related request, and that 831 of those conversations contain a code editing request. We found this number by

searching for markdown-formatted code blocks in the conversations, therefore the actual number of code-related requests is likely higher. We analyzed a subset of code editing requests to understand the types of requests humans make to LLMs. We find that almost all of the requests are of the "lazy" kind that we include in CANITEDIT. We provide two examples of human editing requests in Figure 16. The first example is a request to refactor a Python code snippet, and the second example is a request to refactor a JavaScript code snippet. As shown, these requests are very informal and direct, and do not provide any information about the desired solution. Other instructions we found that we think exemplify the type of instructions humans give to LLMs include:

- *Please change use scrappy instead request.*

- *change this code to python*

- *Can you change above code to not use histogram but use two for loops to create the histogram?*

- *Very cool. Now change it so that it compresses each file using lz4 and saves it to a file with the same name and extension, + ".lz4"*

### E.2 Model-Generated Instructions for Editing Code

```
1   def putMarbles(weights: List[int], k: int) -> int:
2       """
3       You have k bags. You are given a 0-indexed integer array weights where weights[i] is the
4       weight of the ith marble. You are also given the integer k.
5       Divide the marbles into the k bags according to the following rules:
6       No bag is empty.
7       If the ith marble and jth marble are in a bag, then all marbles with an index between the
8       ith and jth indices should also be in that same bag.
9       If a bag consists of all the marbles with an index from i to j inclusively, then the cost
10      of the bag is weights[i] + weights[j].
11      The score after distributing the marbles is the sum of the costs of all the k bags.
12      Return the difference between the maximum and minimum scores among marble distributions.
13      """
14      # code omitted for brevity
15
16  # PASSING TESTS
17
18  # none of the tests are passing
19
20  # FAILING TESTS
21
22  assert putMarbles([1, 3, 5, 1], 2) == 4 # actual output: 6
23  assert putMarbles([1, 3], 2) == 0 # actual output: inf
```

(a) **Instruction:** The implementation failed 2 out of the test cases provided. The issue lies in the calculation of the difference between the maximum and minimum scores among marble distributions. The function returns the difference between the score for k bags and the score for 1 bag, which is not the correct calculation for the problem statement. To fix this issue, we need to find the maximum and minimum scores among all possible distributions and then return the difference between them. This can be achieved by iterating through all possible distributions and keeping track of the maximum and minimum scores, and then returning their difference.

Figure 17: An example of a model-generated instruction for code editing. The instruction is generated by GPT-4 using the Reflexion algorithm (Shinn et al., 2023), by making the model reflect on unit test failures. The problem is from the LeetCode Hard problem set.

This section provides an example of code editing guided by instructions generated by GPT-4 using the Reflexion algorithm. Reflexion is a versatile algorithm developed for enhancing model output through environmental feedback, as detailed in Shinn et al. (2023). While its application extends across various tasks, including reasoning and decision-making, its utility in program synthesis is particularly notable. The process starts with generating unit tests for a program given its natural language description, followed by the creation and evaluation of a candidate program against these tests. If the program fails, Reflexion induces the model to produce a reflection, identifying potential errors and suggesting corrections.

This reflection serves as an instruction for modifying the failing program, which are both provided to the model to edit the failing program into a new candidate, iterating until it passes all tests or a predetermined stop condition is reached.

We provide an example of a model-generated instruction for code editing in Figure 17, where the model was tasked with addressing a problem from the LeetCodeHardGym problem set (Shinn et al., 2023). The instruction, precise and detailed, pinpoints the specific issue in the function's logic and suggests a clear approach for rectification. It emphasizes iterating through marble distributions to calculate the maximum and minimum scores, a method not implemented in the original code. This example showcases how Reflexion can guide models to not only identify errors in logic but also propose viable solutions. This kind of guided instruction is useful for enhancing the accuracy and efficiency of models in complex code editing tasks; however, it is important to note that the instruction is not a complete solution, and that these models may produce misleading or incorrect instructions. The instruction is quite verbose compared to the human examples shown in Figure 16, and it is unclear how humans would interact with such an instruction, as this amount of detail is not necessary for the task at hand.

