# OpenReview forum: "Can It Edit? Evaluating the Ability of Large Language Models to Follow Code Editing Instructions"
_colmweb.org/COLM/2024/Conference — COLM_

### Official Review · Reviewer_Xyri · 2024-04-27

**Rating:** 7
**Confidence:** 5
**Ethics Flag:** 1

**Summary:**

This paper proposes CanItEdit, a hand-curated evaluation benchmark consisting of 105 Python code editing problems, spanning three different categories: corrective edits, adaptive edits, and perfective edits. Each problem comes with two different types of natural language instructions: descriptive (entailing specific details for more explicit guidance) and lazy (minimal instructions that more closely resemble typical user queries). They further construct a training set for this task by filtering the CommitPackFT dataset and also mining additional examples from GitHub commits, where the instructions correspond to commit messages. They demonstrate out-of-the-box, open-source models (CodeLlama, Mixtral, DeepSeekCoder, StarCoder, OctoCoder) fall behind closed models (GPT-3.5-Turbo, GPT-4) by significant margins on this benchmark. By fine-tuning DeepSeekCoder-Base on their new training set, they observe improvements which appear to reduce the gap to a certain extent.

**Questions To Authors:**

- Please provide more details about the dataset creation procedure. For instance, how were the annotators recruited? What were the exact instructions provided to them? It is not clear how the annotators come up with problems. Are they from scratch or is a starting point provided to them? If a starting point is provided, how is it derived? Do these annotators write both the descriptive and lazy instructions. Please explain in more detail the following claim: "Upon completion of a problem, the lead generated sample completions to ensure that the failures and successes were reasonable and consistent with the problem's intent."
- A common use case is updating documentation (e.g., comments, README); however, it is not clear how such documentation edits can be evaluated with test cases. Does this evaluation benchmark include such edits?
- The following claim is made in the paper: "GPT-4, despite not being specifically trained on code-related tasks, surpasses…" Could you explain how you came to this conclusion? It does seem that GPT-4 was trained on a significant amount of code, including come commits data.
- It's not clear why Table 4 was not presented like Table 3, with results for "Descriptive" and "Lazy" shown separately. Do the ablations perform differently for these different use cases?
- Despite motivating the different categories of edits, the main paper does not dive into this at all. I see that some additional results are provided in Table 6 of the Appendix, but this does not include all models in Table 3. Could you share details on whether the same trends are there across these categories for all the models presented in Table 3?

**Reasons To Accept:**

- The authors propose an evaluation set that captures diverse use cases (e.g., editing based on detailed specifications like the descriptive instruction versus editing based on brief user queries like lazy instruction) as well as many different categories of edits: corrective edits (35 examples), adaptive edits (35 examples), and perfective edits (35 examples). I believe this is a novel and very useful contribution to the community as it allows more fine-grained evaluation of LLMs with respect to code editing tasks.
- While I would like to get some more clarity on the details behind this dataset creation procedure (see below), the benchmark is hand-curated, making it likely a very high-quality evaluation set, relative to the few existing code editing datasets. This makes it a very valuable resource for the research community.
- The authors defined a new metric ExcessCode which I believe is very clever and useful. Benchmarks like SWEBench which rely on execution metrics only do not penalize models for making unnecessary changes to the code. I think this metric is also an interesting contribution to the community.
- The experiments are very comprehensive, spanning many different models and dataset ablations. This has highlighted important results with respect to the effect of model size on pass@1 and ExcessCode, and the gap between open and closed source models on this task.

**Reasons To Reject:**

- Very minimal details about the dataset creation procedure are provided. (Please see questions below). This makes it very difficult to understand how the dataset was constructed and consequently assess its quality.
- The dataset is Python-only and does not allow evaluating across different programming languages.

Missing references:
- Automating Code Review Activities by Large-Scale Pre-training (https://arxiv.org/abs/2203.09095)
- CoditT5: Pretraining for Source Code and Natural Language Editing (https://arxiv.org/abs/2208.05446)
- Top Leaderboard Ranking = Top Coding Proficiency, Always? EvoEval: Evolving Coding Benchmarks via LLM (https://arxiv.org/abs/2403.19114)
- CodeEditorBench: Evaluating Code Editing Capability of Large Language Models (https://arxiv.org/abs/2404.03543). (Note: This can be considered contemporary work and I have just included it for reference.)

---

> ### Author Rebuttal · Authors · 2024-05-29
>
> Thank you for the insightful review and suggestions on our work.
>
> > Missing references
>
> We will cite all of these and acknowledge Guo et al. as concurrent work.
>
> > Please provide more details about the dataset creation procedure.
>
> The annotators were recruited through academic networks, these included researchers and engineers in formal methods, programming languages, AI, astrophysics, and mathematics, all of whom use Python heavily in their work. They were encouraged to come up with problems covering a range of difficulty to differentiate models that are very strong at code editing tasks from ones that are not. The annotators were provided with the breakdown of questions from the outset, and each was told what topic and change type to focus on. For each problem, the annotator wrote both the lazy and descriptive text, which later required approval by the lead. There was no starter code for the annotators, however, the lead gave annotators example problems to look at before they began. They were taught how to write the questions unambiguously to give the model a fair chance of passing the unit tests. For example, an ambiguous instruction may cause the model to implement a function body correctly but with a name that was not consistent with the unit test. The project lead reviewed all instructions and several model completions to make sure this wasn't the case.
>
> > Does this evaluation benchmark include [documentation-related] edits?
>
> This benchmark does not deal with updates in documentation such as comments or README's, since they can not straightforwardly be tested with unit tests.
>
> > "GPT-4…" Could you explain how you came to this conclusion?
>
> It is true that GPT-4 is certainly trained on data involving code. We meant is that it's not trained primarily on code, which is the case for models such as StarCoder. We will clarify in the revised version.
>
> > It's not clear why Table 4 was not presented like Table 3
>
> In our ablations we were interested in identifying the overall most effective training configuration, thus we displayed the overall pass@1 and ExcessCode scores. We will include the table with lazy and descriptive columns in the revised version.
>
> > Table 6 does not include all models in Table 3.
>
> We provided only EditCoder and OpenAI models in Table 6 to compare the best open models to closed models. We agree that showing all models would be insightful and will provide the full table in the main body for the revision.

---

> > ### Comment · Reviewer_Xyri · 2024-06-04
> > **Response to rebuttal**
> >
> > Thanks for your detailed response. I will be keeping my score the same.

---

### Official Review · Reviewer_R93Z · 2024-05-08

**Rating:** 6
**Confidence:** 3
**Ethics Flag:** 1

**Summary:**

This paper proposes a new benchmark to evaluate the instruction following ability of Large Language Models (LLMs) in code editing tasks. The evaluation exposes a significant gap between the capabilities of state-of-the-art open and closed models. For example, even GPT-3.5-Turbo is better than the best open-sourced model at code editing tasks. This paper also collects a new dataset named CANITEDIT comprising 105 meticulously constructed Python code editing challenges.

**Reasons To Accept:**

1. This paper proposes a new benchmark containing 105 Python code editing challenges to evaluate the ability of LLMs in code editing tasks. The topic is interesting and the contributions are significant.
2. Several popular open-sourced code LLMs such as CodeLlama and DeepSeekCoder have been evaluated and compared with strong closed-sourced LLMs such as GPT-4.
3. The presentation is clear and the writing is easy to follow.

**Reasons To Reject:**

1. The evaluation metric is limited as there are only pass@k and ExcessCode in metrics. Authors might propose some refined assessment techniques for code editing tasks or add human evaluation.
2. Similar contributions with Guo et al.[1]. Authors might consider to compare with this work if possible.
3. The code languages types are limited since there is only Python language.


[1] Guo et al., CodeEditorBench: Evaluating Code Editing Capability of Large Language Models.

---

> ### Author Rebuttal · Authors · 2024-05-29
>
> Thank you for the insightful review and suggestions on our work.
>
> > The evaluation metric is limited as there are only pass@k and ExcessCode in metrics. Authors might propose some refined assessment techniques for code editing tasks or add human evaluation.
>
> We note that the outputs of certain models did go through human inspection to ensure that all questions were unambiguous. During this process, we were able to get a qualitative understanding of how our models were performing. We provide some of these completions in Appendix D. We would be happy to include a more comprehensive human evaluation in the camera-ready if needed for acceptance.
>
> > Similar contributions with Guo et al.[1]. Authors might consider to compare with this work if possible.
>
> Please note that Guo et al. appeared on ArXiv in April 2024, after the COLM submission deadline. Guo et al. sources problems from popular competitive programming websites, such as LeetCode and CodeForces, thus focusing on data structures and algorithms, while our benchmark encompasses several other topic categories. Additionally, our dataset features unique sets of dual (lazy and descriptive) instructions for each problem, unlike the hardcoded prompts used in their edit categories. We will acknowledge Guo et al. as concurrent work in our revision.
>
> > The code languages types are limited since there is only Python language.
>
> We acknowledge that there are inherent limitations to a benchmark of problems written only in Python. However, mechanically translating Python benchmarks to other languages is an active research topic [1, 2, 3]. We identify expanding our benchmark to other languages as an opportunity for future work.
>
> [1] Cassano et al., MultiPL-E: A Scalable and Extensible Approach to Benchmarking Neural Code Generation
>
> [2] Orlanski et al., Measuring The Impact Of Programming Language Distribution
>
> [3] Athiwaratkun et al., Multi-lingual Evaluation of Code Generation Models

---

### Official Review · Reviewer_b7CL · 2024-05-10

**Rating:** 7
**Confidence:** 4
**Ethics Flag:** 1

**Summary:**

The paper proposes a novel benchmark called CANITEDIT, which is designed to evaluate the instructional code editing skills of Code LLMs. It comprises 105 manual code editing tasks, each accompanied by two sets of natural language instructions and a comprehensive test suite for validation. Additionally, it presents  a specially tailored training dataset for code editing. Through extensive experiments, the work highlights a significant performance contrast between closed and open models among state-of-the-art Code LLMs. Moreover, the presented training dataset has been shown to be effective in improving code editting capabilities of models at various sizes.

**Questions To Authors:**

See the comments above.

**Reasons To Accept:**

* The proposed benchmark is novel. It expands the evaluation scope of code LLMs to include instructional code editing skills, an often overlooked aspect in their assessment.
* The evaluation is comprehensive and provides insightful findings. This work underscores a significant performance contrast between closed and open code LLMs. Moreover, experimental results verify the efficacy of the curated training dataset in enhancing code editing capabilities across diverse model sizes.
* This paper is well organized and presented.

**Reasons To Reject:**

* A potential concern arises from the limited scope of the benchmark, which encompasses only 105 manually crafted instructional code editing problems. This constraint could result in a restricted coverage of domains within the evaluation.

---

> ### Author Rebuttal · Authors · 2024-05-29
>
> Thank you for the insightful review and suggestions on our work.
>
> > A potential concern arises from the limited scope of the benchmark, which encompasses only 105 manually crafted instructional code editing problems. This constraint could result in a restricted coverage of domains within the evaluation.
>
> We acknowledge that CanItEdit only contains 105 tasks, but we point out the following:
>
> 1. We provide two variants of task specifications per task, effectively creating 210 problems of varying difficulty.
> 2. Several other notable code generation benchmarks are composed of between 100-200 tasks, such as HumanEval [1] and LeetcodeHardGym [2]
> 3. Among the 210 problems, CanItEdit covers over 5 broad categories topics, with 3 different expected change types.
>
> We hope that the high quality and diverse range of tasks within our dataset effectively mitigate limitations posed by its modest size, ensuring robust and comprehensive evaluations.
>
> [1] Chen et al., Evaluating Large Language Models Trained on Code
>
> [2] Shinn et al., Reflexion: Language Agents with Verbal Reinforcement Learning

---

### Official Review · Reviewer_NZ9v · 2024-05-16

**Rating:** 7
**Confidence:** 4
**Ethics Flag:** 1

**Summary:**

This paper proposes a benchmark to evaluate the instruction-following code editing capability of selected baselines.

- The authors include several kinds of editing instructions in the design.
- The authors provide an instruction-tuning training dataset bundle specifically for code editing enhancement.

**Questions To Authors:**

"These suites were designed to rigorously evaluate whether the ’after’ segment met the problem requirements while ensuring the ’before’ code did not." -- if all the "before" segments do not met the requirements, do they still fall into any of the three task categories?

Can the authors provide clarification on the "relatively small" and "quite low" from "The mean Levenshtein distance between the ‘before’ and ‘after’ code segments is 197.1 characters, indicating that the changes are relatively small. We also analyze the distribution of the commit message lengths, and find that the mean token count is 10.1, which is quite low."?

Is there any overlap between the two training sets?

**Reasons To Accept:**

Compare to the portion of editing tasks in code-related conversations (19%) , the research on such specific kind of task is absent for LLMs. We need to study more about the LLMs on the capability of manipulating existing codes, distinguished from direct code generation from natural languages.

- The authors provide a compact hand-crafted code editing instruction dataset for benchmarking. The curated commit-based fine-tuning dataset would be a useful resource for future research.
- The proposed benchmark adapts the existing  *corrective*, *perfective*, and *adaptive* editing task classification system, and further uses a *descriptive-lazy* instruction categorisation dimension, which is a concise fit for LLM research. Based on which, there are in-depth analyses of the results appended.

**Reasons To Reject:**

- **Quantified correlation with the code generation capability.**  Although this paper focuses on the code editing task, the authors should also include explore the correlation between the general code generation and editing performances of the baselines in the main text, given that there is assumption that the former could be the basis of the latter.
- **Lacking of human evaluation.** It would be better if the authors could provide the performance of human annotators in the benchmark as a guideline for the (half) open-ended problems.

---

> ### Author Rebuttal · Authors · 2024-05-29
>
> Thank you for the insightful review and suggestions on our work.
>
> > Quantified correlation with the code generation capability.
>
> We agree that code generation and code editing capability are  correlated. It's possible that if a model exhibits strong code generation capabilities, it will also show strong performance on our benchmark. In Appendix A.4 we provide a table with results of some of our models on the HumanEvalPack benchmark, which includes a code generation split (Synthetize). We compute the Pearson correlation coefficient between the Synthetize scores and the average between the Lazy and Descriptive scores on CanItEdit. We find a coefficient of 0.661, suggesting a moderate positive correlation. We will add this result to the revised version of the paper.
>
> > Lacking of human evaluation. It would be better if the authors could provide the performance of human annotators in the benchmark as a guideline for the (half) open-ended problems.
>
> We would be happy to include this in the camera-ready if needed for acceptance.
>
> > if all the "before" segments do not met the requirements, do they still fall into any of the three task categories?
>
> We categorized problems as corrective, perfective, or adaptive based on the changes applied in the ground truth solution.
> A model may produce an edit that does not conform to the labeled changed type. However, such an edit likely won't pass the test cases, as we have built them to reflect the correct change kind. We will clarify in the revised version.
>
> > Can the authors provide clarification on the "relatively small" and "quite low" from "The mean Levenshtein distance between the ‘before’ and ‘after’ code segments is 197.1 characters, indicating that the changes are relatively small. We also analyze the distribution of the commit message lengths, and find that the mean token count is 10.1, which is quite low."?
>
> We believe these numbers are small with respect to the ones found in our benchmark. In our benchmark, the mean Levenshtein distance is 302.1, while the mean token count is 81.7 for descriptive instructions and 35.6 for lazy instructions. We will clarify in the revised version.
>
> > Is there any overlap between the two training sets?
>
> We have found 4 overlapping items between EditPackFT and Commits2023FT by finding exact matches between the before, instruction, and after chunks. These overlapping items are filtered out by the MinHash+LSH deduplication process when the two datasets are combined.

---

### Decision · Program_Chairs · 2024-07-10

**Decision:**

Accept

**Comment:**

This work introduces a code-editing benchmark and conducts evaluation for many open-sourced and commercial code models.
Finetuning experiments are also presented in the paper, showing that they are helpful in improving performance across the board.
While the size (Reviewer b7CL) and language coverage (Reviewers R93Z and Xyri) of the introduced benchmark are limited, the paper presents a first-step effort toward evaluating LLM code editing. The paper is well-written, and the results are clearly presented.

Reviewers NZ9v and R93Z have suggested adding human evaluation. I agree with them that either comparing human and model behaviors or having human feedback would be an interesting direction that lies in the intersection among NLP, program synthesis, psychology, and HCI; however, since the task itself is well-defined and can be evaluated by comparing execution results, I don't think this paper needs to compare the model behaviors with humans for acceptance.

I agree with Reviewer Xyri that more details about the dataset curation would be helpful. Please incorporate your response into the next version of the paper.

[comments form the PCs] It's important to follow the AC's recommendation regarding dataset curation details.

[At least one review was discounted during the decision process due to quality]